# HIV-1 envelope glycoprotein modulates CXCR4 clustering and dynamics on the T cell membrane

Adriana Quijada-Freire[1], César A Santiago[2], Eva M García-Cuesta[1], Blanca Soler Palacios[1], Rosa Ayala-Bueno[1], Sofia R Gardeta[1], Enara San Sebastian[3], Eva Armendariz-Burgoa[4], Maria Carmen Puertas[4], Ricardo Villares[1], Urtzi Garaigorta[3], Luis Ignacio González-Granado[5,6], Jose Miguel Rodriguez Frade[1], Jakub Chojnacki[4,7,8], Javier Martinez-Picado[4,7,9,10], Mario Mellado[1]*

[1]Chemokine Signaling group, Department of Immunology and Oncology, Centro Nacional de Biotecnología/CSIC, Madrid, Spain; [2]X-ray Crystallography Unit, Department of Macromolecules Structure, Centro Nacional de Biotecnología/CSIC, Madrid, Spain; [3]Departamento de Biología Molecular y Celular, Centro Nacional de Biotecnología/CSIC, Madrid, Spain; [4]IrsiCaixa, Badalona, Spain; [5]12 de Octubre Health Research Institute (imas12), Madrid, Spain; [6]Department of Public Health School of Medicine, School of Medicine Universidad Complutense de Madrid, Madrid, Spain; [7]CIBERINFEC, Madrid, Spain; [8]Germans Trias i Pujol Research Institute (IGTP), Badalona, Spain; [9]University of Vic-Central University of Catalonia, Vic, Spain; [10]Catalan Institution for Research and Advanced Studies (ICREA), Barcelona, Spain

**\*For correspondence:**
mmellado@cnb.csic.es

## eLife Assessment

This study provides **valuable** insights into how HIV-1 Env modulates the nanoscale organization and dynamics of the CXCR4 co-receptor on T cells, using quantitative imaging and functional approaches, the authors present **convincing** evidence that gp120 engagement promotes CD4-dependent clustering and altered mobility of CXCR4, distinct from the effects of the natural ligand CXCL12. Some concerns were raised regarding the interpretation of the single-particle tracking analyses, and additional clarification or analysis may help strengthen the conclusions. The physiological relevance of the findings could be further enhanced by validation with infectious virus and by more clearly integrating the CXCR4R334X mutant observations into the central mechanistic narrative. The work will be of interest to researchers studying HIV entry and membrane receptor organization. [Editors' note: this paper was reviewed by Review Commons.]

**Abstract** HIV-1 entry into susceptible cells requires the dynamic interaction of its envelope (Env) glycoprotein with the host cell receptor CD4 and a co-receptor, either CCR5 or CXCR4. While the core molecular mechanisms driving Env-receptor interactions and subsequent membrane fusion are well characterized, the precise nanoscale spatial reorganization of these co-receptors at the viral binding site remains poorly defined. In this study, we employed single-particle tracking total internal reflection fluorescence (SPT-TIRF) microscopy to quantitatively analyze nanoscale organizational changes of CXCR4 on the surface of human CD4+ T cells following binding by X4-tropic HIV-1. Our data reveal that both recombinant X4-gp120 and virus-like particles expressing physiological levels of X4 Env proteins (gp120 and gp41) promote CXCR4 clustering, a phenomenon linked to cell infection. Furthermore, these ligands induced oligomerization of CXCR4R334X, a naturally occurring mutant

associated with WHIM syndrome that supports HIV-1 infection, but fails to oligomerize in response to CXCL12. Our findings establish a link between CXCR4 clustering and HIV-1 infection, enhancing our understanding of the initial events in viral attachment and entry. These results further suggest that HIV-1 depends on a specific spatial arrangement of co-receptors, distinct from that induced by their natural chemokine ligands, highlighting the critical role of cell-surface receptor spatial organization in dictating cellular function.

## Introduction

HIV-1 infects immune cells through a dynamic interaction between its envelope glycoprotein complex (Env), composed of gp41 and gp120 trimers (*Kowalski et al., 1987*; *Lu et al., 1995*), and two receptors on target cells: the primary receptor CD4, and a co-receptor, either CCR5 or CXCR4. These co-receptors are critical for viral entry and determine viral tropism. R5 strains (M-tropic) infect primary macrophages and certain memory CD4$^+$ T cells by binding to CD4 and CCR5 (*Choe et al., 1996*), while X4 strains (T-tropic) infect primary CD4$^+$ T cells by binding to CD4 and CXCR4 (*Feng et al., 1996*).

R5-tropic viruses are typically the primary mode of transmission in humans and remain prevalent throughout the infection (*Wolinsky et al., 1996*). However, in many infected individuals, the virus evolves its tropism over time, progressing from R5-tropic to dual-tropic (R5/X4), and ultimately to X4-tropic as the infection advances (*Lu et al., 1997*). This shift in tropism is associated with a more rapid decline in CD4$^+$ T cell counts (*Connor et al., 1993*). Furthermore, CXCR4 has been implicated in the infection of hematopoietic progenitor cells, potentially contributing to the establishment of long-lasting latent HIV-1 reservoirs (*Carter et al., 2011b*).

The fusion of viral and cellular membranes is initiated by the binding of gp120 to CD4 (*Landau et al., 1988*; *Arthos et al., 1989*), which triggers a conformational change in gp120 that facilitates co-receptor engagement (*Zhang et al., 1999*). These structural changes expose the N-terminal hydrophobic fusion peptide of gp41, which inserts into the cell membrane (*Doms and Moore, 2000*) to establish a fusion pore and release the viral contents into the target cell.

Numerous structural and biophysical studies have examined the HIV-1 infection process, successfully clarifying the structural requirements and conformational changes in gp120 and gp41 that enable membrane fusion and viral entry (*Chojnacki et al., 2012*; *Buzon et al., 2010*; *Pancera et al., 2010*; *Garg et al., 2011*). Early investigations into host cell components revealed that HIV-1 infection induces the redistribution of cell-surface CD4 and co-receptors to the sites of viral attachment (*Barrero-Villar et al., 2009*; *Jiménez-Baranda et al., 2007*). More recent evidence, obtained through super-resolution microscopy, demonstrates that HIV-1 binding triggers CD4 clustering, a phenomenon also observed (albeit to a lesser degree) following stimulation with gp120 alone (*Yuan et al., 2021*). However, less is known about which specific co-receptors are also recruited to the virus particles bound to the cell surface. It is known that gp120 can mediate the association of CD4 with both CXCR4 and CCR5 (*Barrero-Villar et al., 2009*; *Lapham et al., 1996*; *Ugolini et al., 1997*). Furthermore, it has been shown that the co-expression of CCR5 modifies the conformation of both CXCR4 homodimers and CD4/CXCR4 heterodimers. This conformational change prevents the binding of gp120 from an X4 HIV-1 strain, specifically gp120$_{IIIB}$, to the resulting CD4/CXCR4/CCR5 complex (*Martínez-Muñoz et al., 2014*).

Large-scale molecular assemblies at the cell membrane, often termed signaling clusters or nanoclusters, are increasingly recognized as key regulators of cell signaling (*Hartman and Groves, 2011*). Evidence indicates that the spatial reorganization and distribution of membrane receptors are key to controlling cell functions. Notably, clustering also appears to be essential for viral infection. For instance, Env–CD4 complexes form clusters and ring-like structures, facilitating closer contact between opposing membranes (*Li et al., 2023*). These clusters often govern a significant portion of the overall signaling process and are frequently associated with the cytoskeleton (*Hartman and Groves, 2011*). For example, CD4 receptors exist in pre-clustered structures that enlarge upon T cell activation, thereby modulating the strength of the signaling response (*Krummel et al., 2000*; *Kao et al., 2008*). Moreover, CXCR4 is organized at the cell membrane as monomers, dimers, and small aggregates (groups of ≥3 receptors) called nanoclusters. The binding of its specific ligand, CXCL12, leads to a decrease in the proportion of monomers/dimers, while simultaneously increasing

the formation of larger nanoclusters. This change in receptor organization consequently alters the lateral mobility of CXCR4 within the cell membrane (*Martínez-Muñoz et al., 2018*). This mechanism is critical for initiating CXCR4 signaling and enabling cells to accurately orient themselves in response to CXCL12 gradients (*García-Cuesta et al., 2022*). A naturally occurring CXCR4 mutant, CXCR4$^{R334X}$, which is responsible for WHIM syndrome, a severe immunological disorder (*Liu et al., 2012*), fails to form these large nanoclusters upon CXCL12 binding. Consequently, cells carrying this mutation lose their ability to properly sense chemoattractant gradients (*García-Cuesta et al., 2022*).

Here, we employed quantitative single-particle tracking in total internal reflection fluorescence (SPT-TIRF) microscopy to directly investigate the spatial arrangement and dynamic activity of CXCR4 upon exposure to HIV-1 glycoproteins. Our findings reveal that both recombinant X4-gp120 and virus-like particles (VLPs) containing a limited number of X4 Env proteins (gp120 and gp41) promote CXCR4 clustering on cells expressing CD4 and CXCR4. Moreover, they trigger the oligomerization of CXCR4$^{R334X}$. These results suggest that, along with CD4 clustering, the conformational changes in chemokine receptors triggered by HIV-1 are essential for cell infection and differ from the effects of the natural ligand, CXCL12. Therefore, CD4/CXCR4 complexes and their interaction sites represent potentially valuable targets for developing novel therapeutic strategies to block HIV-1 entry.

## Results
### Recombinant gp120-mediated CXCR4 clustering requires CD4 expression

To evaluate the role of HIV-1 in modulating CXCR4 dynamics at the cell membrane, we first synthesized a recombinant X4 HIV-1 gp120 (X4-gp120) with a C-terminal histidine tag to facilitate subsequent detection. We generated a HEK-293T cell line that constitutively secretes the recombinant glycoprotein. The secreted X4-gp120 was isolated from cell culture supernatants using a simple, rapid, non-denaturing, and efficient purification procedure involving Ni-NTA agarose chromatography and gel filtration chromatography. The purity of the isolated glycoprotein was confirmed by SDS-PAGE and western blotting, using commercial gp120 as a control and specific antibodies (*Figure 1—figure supplement 1A and B*; *Figure 1—figure supplement 1—source data 1–4*). The X4-gp120 specifically bound to Jurkat cells expressing CD4, regardless of the presence or absence of CXCR4 (JKCD4$^+$CXCR4$^+$ and JKCD4$^+$CXCR4$^-$, respectively; *Figure 1—figure supplement 1C*). By contrast, X4-gp120 did not bind to Daudi cells, which express only CXCR4 (*Figure 1—figure supplement 1D*). In this latter case, X4-gp120 binding became possible when the recombinant protein was pre-incubated with soluble CD4. As a control, no binding of soluble CD4 was detected in the absence of X4-gp120 (*Figure 1—figure supplement 1D*). These data indicate that our recombinant gp120 is a X4-tropic gp120 capable of binding Jurkat cells expressing CD4.

Engagement of the HIV-1 Env with CD4 and/or a chemokine co-receptor activates several signal transduction pathways (*Balabanian et al., 2004*; *Arthos et al., 2000*). For instance, gp120 binding to CD4 leads to the phosphorylation of the receptor tyrosine kinase p56Lck, which then activates the Raf/MEK/ERK and phosphatidylinositol 3-kinase (PI3K) pathways (*Briand et al., 1997*). PI3K is also activated through chemokine receptor engagement (*Curnock et al., 2002*), and only viruses capable of inducing signaling *via* these receptors can establish productive infections (*Arthos et al., 2000*). Jurkat cells and primary CD4$^+$ T blasts were stimulated with X4-gp120, then lysed and analyzed by western blotting. The results showed that Akt, ERK1/2, and Lck were rapidly phosphorylated following X4-gp120 activation (*Figure 1—figure supplement 2*; *Figure 1—figure supplement 2—source data 1–4*) in both cell types. Furthermore, it is known that the interaction of gp120 with CXCR4 influences the actin cytoskeleton (*Yoder et al., 2008*). Cytochalasin D, an inhibitor of F-actin polymerization, has been shown to inhibit viral entry into PBMCs (*Iyengar et al., 1998*), and to affect the formation of HIV-1 reverse transcriptase products in HeLa cells (*Bukrinskaya, 2004*). We thus evaluated whether X4-gp120 reorganized the actin cytoskeleton of the target cells. Using artificial lipid bilayers coated with ICAM-1, we observed that both CXCL12 and X4-gp120 induced the polarization of primary CD4$^+$ T blasts and, furthermore, enhanced cell adhesion (*Figure 1—figure supplement 3A and B*). However, only CXCL12 triggered significant cell migration (*Figure 1—figure supplement 3C*). These findings confirmed that X4-gp120 bound CXCR4 in the presence of CD4 and was fully functional on both primary CD4$^+$ T blasts and Jurkat cells.

The actin cytoskeleton acts as a physical barrier, influencing the compartmentalization of the plasma membrane and the dynamics of membrane proteins (*Plowman et al., 2005*; *Torreno-Pina et al., 2016*). Consequently, the dynamic nature of actin not only defines cell shape during migration, but also affects membrane organization of both CD4 (*Pereira et al., 2019*) and CXCR4 (*Martínez-Muñoz et al., 2018*). Next, we transiently transfected JKCD4$^+$X4$^-$ cells, which express endogenous CD4, with CXCR4-AcGFP. We then used SPT in TIRF-M mode to observe individual molecules within the plasma membrane, allowing us to determine the effect of X4-gp120 on CXCR4 dynamics and stoichiometry (*Figure 1—videos 1–3*). Consistent with previous observations (*Martínez-Muñoz et al., 2018*; *García-Cuesta et al., 2022*), our analysis of CXCR4 dynamics in unstimulated cells revealed that the majority of CXCR4 particles were mobile (~87%) (*Figure 1A*), exhibiting a median short time-lag diffusion coefficient ($D_{1-4}$) of 0.017 µm$^2$ s$^{-1}$ (*Figure 1B*). Upon stimulation, both CXCL12 and X4-gp120 significantly reduced the overall receptor diffusivity (CXCL12, median $D_{1-4}$ = 0.007 µm$^2$ s$^{-1}$; X4-gp120, median $D_{1-4}$ = 0.009 µm$^2$ s$^{-1}$; *Figure 1B*), and increased the percentage of immobile particles from ~13% (basal) to ~20% (CXCL12), and to ~18% (X4-gp120; *Figure 1A*). Mobile particles exhibited distinct diffusion profiles, derived from mean square displacement (MSD) plots (*Manzo and Garcia-Parajo, 2015*), and were further classified based on motion using the moment scaling spectrum (*Ewers et al., 2005*). For most mobile particles (~90% in unstimulated cells, ~79% in CXCL12-activated cells, and ~75% in X4-gp120 stimulated cells), diffusion was confined (*Figure 1—figure supplement 4*). To quantify the number of receptors in individual trajectories, we measured the average fluorescence intensity during the initial 20 frames of each trajectory and used the intensity of the monomeric protein CD86-AcGFP as a reference (*García-Cuesta et al., 2022*; *Calebiro et al., 2013*; *Figure 1—figure supplement 5*). In unstimulated cells, we found a predominance of CXCR4 monomers and dimers (~98%), with only a minor fraction of oligomers, complexes containing more than three receptors (~2%). Upon the addition of saturating X4-gp120 concentrations, we observed a significant reduction in the percentage of monomers and dimers (~82%) and a corresponding increase in nanoclusters composed of ≥3 receptors/particle (~18%) (*Figure 1C*). This observation aligns with previous findings for CXCL12 (*Martínez-Muñoz et al., 2018*). As a control, stimulation with CXCL12 resulted in a larger percentage of these nanoclusters (~26%; *Figure 1C*). These data correlated with the higher MSI values observed after cell activation (basal 933 a.u.; CXCL12 2,105 a.u., X4-gp120 1,738 a.u.; *Figure 1D*). Collectively, these results indicate that X4-gp120 triggers CXCR4 nanoclustering, although to a lesser extent than CXCL12.

## gp120-expressing virus-like particles mediate CXCR4 clustering

Recombinant X4-gp120 alone does not fully replicate the function of HIV-1 Env. Previous studies have shown that the Env consists of gp120 trimers that redistribute and cluster on the surface of virions following proteolytic Gag cleavage during maturation (*Chojnacki et al., 2017*). Considering the low number of Env trimers on natural HIV virions (*Hart et al., 1993*; *Zhu et al., 2003*; *Zhu et al., 2006*), this clustering is crucial for establishing multiple receptor interactions necessary for virus entry. To mimic the behavior of the virus, we prepared VLPs containing the X4 HIV-1 Env. HEK-293T cells were transiently transfected with pHXB2env, psPAX2, and, when required, a pLentiGFP plasmid that encodes a GFP reporter gene flanked by LTR regions. In this latter case, we generated LVPs because these budding particles contained genetic material and could transduce target cells. Supernatants were collected 48 hr post-transfection, and VLPs were purified by ultracentrifugation and resuspended in PBS. The structural integrity of the VLPs was confirmed using negative-stain electron microscopy (*Figure 2—figure supplement 1A*). Western blot analysis of the samples, culture media, and clarified supernatants confirmed the presence of both gp120 and p24 in VLPs and LVPs (*Figure 2—figure supplement 1B*; *Figure 2—figure supplement 1—source data 1 and 2*). The gp120-containing viral particles bound soluble CD4 (*Figure 2—figure supplement 1C*), and the corresponding LVPs successfully infected HEK-293CD4 cells, as demonstrated in transduction assays (*Figure 2—figure supplement 1D and E*). LVPs containing VSVG were utilized as a positive control in these functional assays (*Figure 2—figure supplement 1D and E*).

Immature HIV-1 particles exhibit reduced entry efficiency (*Murakami et al., 2004*; *Wyma et al., 2004*). This effect may stem from the rigidity of the immature Gag lattice beneath the viral membrane, which hinders membrane fusion (*Rauh et al., 2005*), interactions between Gag and Env glycoproteins, limiting the lateral mobility of the sparsely distributed Env trimers and, consequently, impairing the

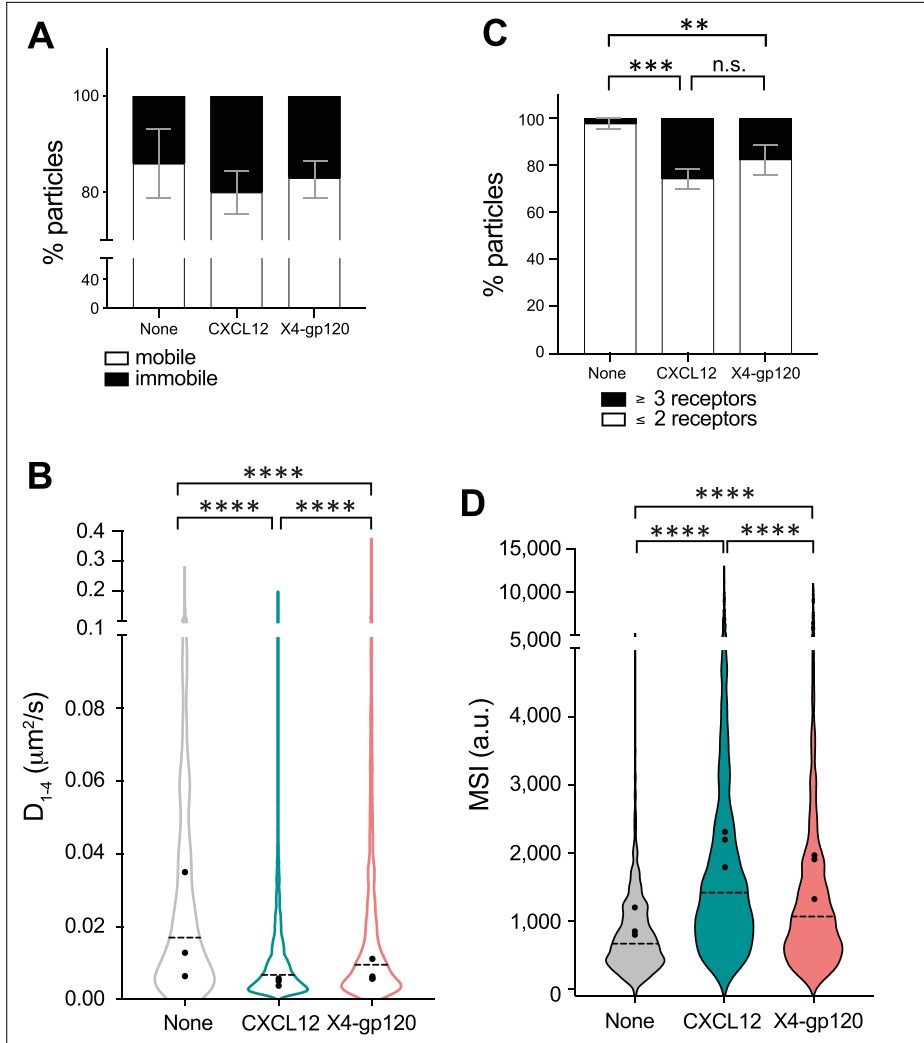

**Figure 1.** X4-gp120 modulates CXCR4 dynamics and nanoclustering. Single-particle tracking analysis of JKCD4+X4- cells transiently transfected with CXCR4-AcGFP on fibronectin (FN)-, FN +CXCL12-, or FN +X4-gp120-coated coverslips (828 particles in 96 cells on FN; 2997 in 95 cells on FN +CXCL12 and 1547 in 91 cells on FN +X4-gp120) n=3. (**A**) Percentage of mobile and immobile CXCR4-AcGFP particles at the membrane of cells treated as indicated. (**B**) Diffusion coefficients ($D_{1-4}$) of mobile particles at the membrane of cells treated as indicated with the median value of each experiment (black circles) and the median of all trajectories (dotted black lines; ****p≤0.0001). (**C**) Frequency of CXCR4-AcGFP particles containing monomers and dimers (≤2) or nanoclusters (≥3), mean ± SD calculated from mean spot intensity (MSI) values of each particle as compared with the value of monomeric CD86-AcGFP (980±86 a.u., **p≤0.01, ***p≤0.001). (**D**) Intensity distribution of individual CXCR4-AcGFP trajectories on unstimulated and CXCL12 or X4-gp120-stimulated cells. Graph shows the distribution of all trajectories, with the mean value of each experiment (black circles) and the median of all trajectories (dotted black lines; n=3; ****p≤0.0001). Statistical significance was determined by two-way ANOVA in panels A and C and by non-parametric Kruskal-Wallis tests followed by Dunn's test for panels **B** and **D**.

The online version of this article includes the following video, source data, and figure supplement(s) for figure 1:

**Figure supplement 1.** Generation of functional recombinant X4-gp120.

**Figure supplement 1—source data 1.** Original files for Coomassie blue-stained polyacrylamide gel for *Figure 1—figure supplement 1A*.

**Figure supplement 1—source data 2.** PDF file containing original Coomassie blue-stained polyacrylamide gel for *Figure 1—figure supplement 1A*.

**Figure supplement 1—source data 3.** Original files for western blot analysis for *Figure 1—figure supplement 1B*.

*Figure 1 continued on next page*

*Figure 1 continued*

**Figure supplement 1—source data 4.** PDF file containing original western blot for *Figure 1—figure supplement 1B*.

**Figure supplement 2.** X4-gp120 activates CD4- and CXCR4-related signaling pathways in target cells.

**Figure supplement 2—source data 1.** Original files for western blot analysis for *Figure 1—figure supplement 2A*.

**Figure supplement 2—source data 2.** PDF file containing original western blot for *Figure 1—figure supplement 2A*.

**Figure supplement 2—source data 3.** Original files for western blot analysis for *Figure 1—figure supplement 2B*.

**Figure supplement 2—source data 4.** PDF file containing original western blot for *Figure 1—figure supplement 2B*.

**Figure supplement 3.** X4-gp120 induces cell polarization.

**Figure supplement 4.** Classification of mobile receptor trajectories.

**Figure supplement 5.** Characterization and calculation of reference parameters for particles intensity, related to TIRF images.

**Figure 1—video 1.** Representative video of CXCR4-AcGFP on live JKCD4+X4- cells treated with DMSO and captured by SPT- TIRF, showing the diffusion of CXCR4 particles (monomers, dimers, and nanoclusters) at steady state (FN).
https://elifesciences.org/articles/110354/figures#fig1video1

**Figure 1—video 2.** Representative video of CXCR4-AcGFP on live JKCD4+X4- cells treated with DMSO and captured by SPT-TIRF, showing the diffusion of CXCR4 particles (monomers, dimers, and nanoclusters) in response to CXCL12.
https://elifesciences.org/articles/110354/figures#fig1video2

**Figure 1—video 3.** Representative video of CXCR4-AcGFP on live JKCD4+X4- cells treated with DMSO and captured by SPT-TIRF, showing the diffusion of CXCR4 particles (monomers, dimers, and nanoclusters) in response to X4-gp120.
https://elifesciences.org/articles/110354/figures#fig1video3

clustering necessary for efficient infection (*Chojnacki et al., 2012*). Therefore, we evaluated the maturation status of the generated VLPs using STED microscopy. We compared the condensation of Gag and the distribution of Env molecules on the surface of the VLPs with those observed on genetically immature particles and integrase-defective NL4-3ΔIN virions, serving as controls (*Chojnacki et al., 2012*). Env proteins were stained with the human mAb 2G12, which specifically recognizes the gp120 domain. To avoid antibody-induced clustering, we used purified Fab fragments of 2G12. The location of individual HIV-1 particles was determined using the human mAb 37G12, which targets the Gag protein and served as a '"counterstain'" reference (*Figure 2A*). By assessing Gag condensation, we estimated that ~75% of the VLPs, both gp120-VLPs and Env(-) VLPs, were mature. These percentages were lower for NL4-3ΔIN virions (~50%) and for immature VLPs (~16%) used as controls (*Figure 2B*). Furthermore, we observed that 26.5% of the gp120-VLPs, 40.2% of the NL4-3ΔIN virions, and 40.5% of the immature VLPs expressed gp120. As a control, the anti-gp120 2G12 Fab did not stain particles lacking the Env (Env(-) VLPs; *Figure 2A and C*), confirming the specificity of the staining. Moreover, STED analysis revealed differences in Env distribution between gp120-VLPs and NL4-3ΔIN virions, as previously observed between mature and immature particles (*Chojnacki et al., 2012*). Specifically, gp120 staining intensity was higher for NL4-3ΔIN particles than for gp120-VLPs (NL4-3ΔIN 1786 a.u. vs. gp120-VLPs 1223 a.u.; *Figure 2D*), suggesting a lower expression of Env proteins in the latter or a lower incorporation of the Env proteins into the VLPs. Analysis of gp120 intensity per particle demonstrated that g120-VLPs had lower levels of gp120/particle than NL4-3ΔIN virions (*Figure 2E*). These data confirmed the mature state of the gp120-VLPs generated and indicated that they contained a reduced number of gp120/particle, similar to or even lower than that found in primary HIV-1 viruses (*Zhu et al., 2006*).

Next, we employed SPT-TIRF to investigate how gp120-VLPs affect CXCR4 dynamics in JKCD4⁺X4⁻ cells that were transiently transfected with CXCR4-AcGFP (*Figure 3—videos 1–3*). Analysis indicated that the gp120-VLPs significantly reduced overall receptor diffusivity (basal, $D_{1-4}$ = 0.025 µm² s⁻¹;

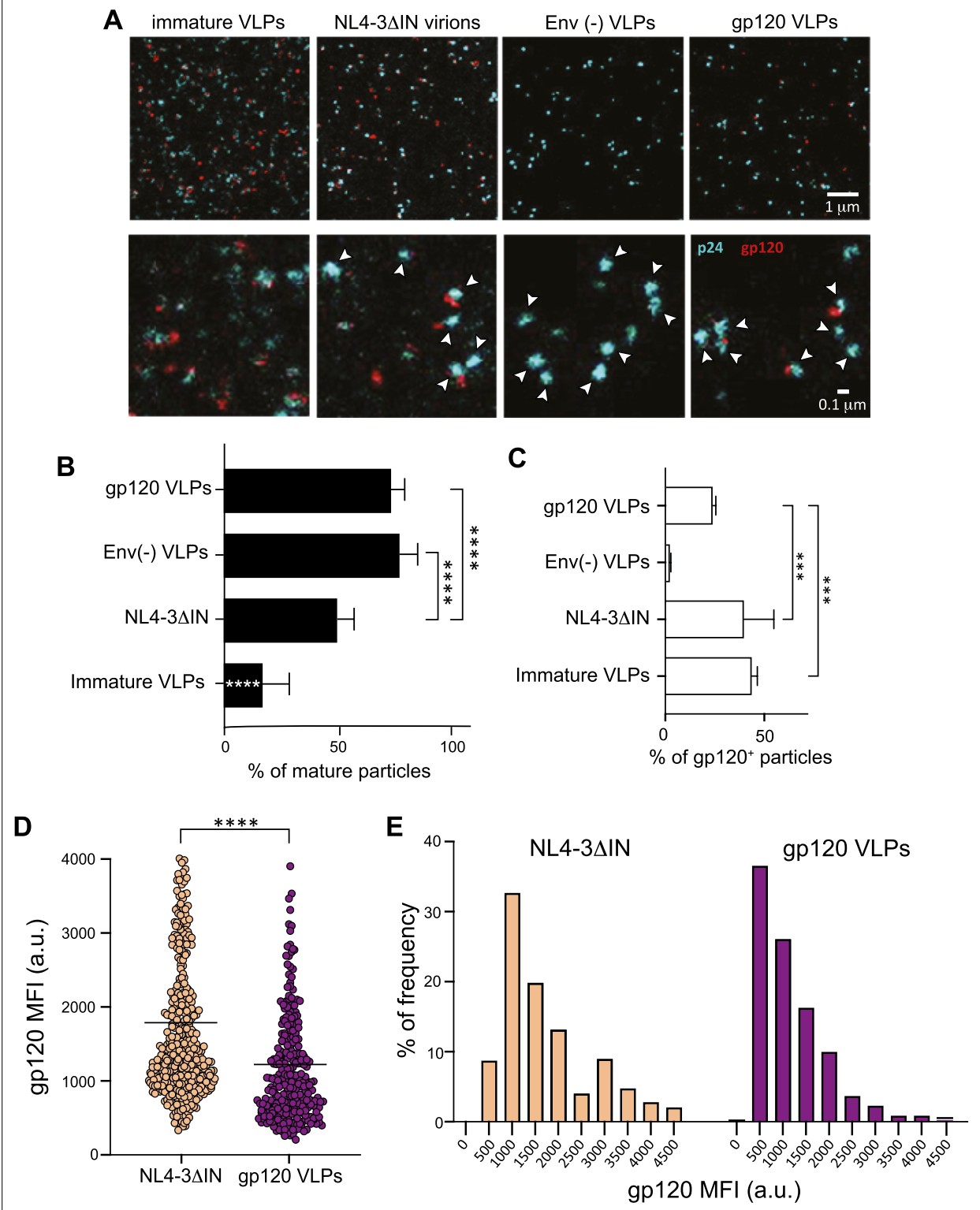

**Figure 2.** gp120-VLPs are mature particles that express a low number of Env trimers. (**A**) Representative images of clarified VLPs visualized by STED microscopy. Upper panels show images of the indicated VLPs stained for Gag p24 (blue) and gp120 (red). Lower panels show ×10 magnification of equivalent images. White arrows indicate mature VLPs (p24 condensation). (**B**) Percentage of mature VLPs, analyzed from the images in (**A**) using TrackAnalyzer in ImageJ, based on p24 intensity and aggregation level (mean ± SD; n=2; ****p≤0.0001; the significance indicated on immature VLPs bar shows the difference with all other conditions). (**C**) Percentage of VLPs expressing gp120 on their surface, as analyzed in ImageJ (mean ± SD; n=2; ***p≤0.001). (**D**) Distribution of gp120 mean fluorescence intensity. Each spot corresponds to the mean fluorescence intensity for each analyzed VLP in

*Figure 2 continued on next page*

*Figure 2 continued*

a.u. The black line represents the mean of all values (****p≤0.0001). (E) Frequency of gp120 intensity/particle. Statistical significance was determined by one-way-ANOVA followed by Tukey's multiple comparisons test in panels B and C and by Mann-Whitney analysis for panel D.

The online version of this article includes the following source data and figure supplement(s) for figure 2:

**Figure supplement 1.** Characterization of gp120-VLPs and LVPs.

**Figure supplement 1—source data 1.** Original files for western blot analysis for *Figure 2—figure supplement 1B*.

**Figure supplement 1—source data 2.** PDF file containing original western blots for *Figure 2—figure supplement 1B*.

Env(-) VLPs, $D_{1-4}$ = 0.017 µm$^2$ s$^{-1}$; gp120-VLPs $D_{1-4}$ = 0.012 µm$^2$ s$^{-1}$; *Figure 3A*). In all the cases, most of the trajectories corresponded to confined movement (*Figure 3—figure supplement 1*). Under these conditions, we found predominantly CXCR4 monomers and dimers at steady-state, and their proportion decreased upon VLP treatment (basal ~99%; Env(-) VLPs ~91%; gp120-VLPs ~71%). Consequently, the percentage of nanoclusters containing ≥3 receptors/particle increased (~1%, basal; ~9%, VLPs; ~29%, gp120-VLPs; *Figure 3B*). These findings were consistent with the MSI values (basal 761 a.u.; Env(-) VLPs 1150 a.u.; gp120-VLPs 1898 a.u.; *Figure 3C*). Further investigations are needed to elucidate the precise mechanism involved in the effect induced by the Env(-) VLPs on the dynamics of the receptors.

Our results indicated that, similar to the effect triggered by soluble X4-gp120, VLPs containing HIV-1 Env also triggered CXCR4 nanoclustering. To understand the role of CXCR4 clustering in HIV-1 infection, we next analyzed the behavior of the WHIM mutant CXCR4, CXCR4$^{R334X}$, which does not oligomerize in the presence of CXCL12 (*García-Cuesta et al., 2022*). CXCR4$^{R334X}$ is a natural mutant of CXCR4 that binds CXCL12 (*Balabanian et al., 2005*; *Hernandez et al., 2003*; *Busillo and Benovic, 2007*), but is not internalized in response to the ligand; in fact, it is a gain-of-function receptor for this ligand (*McDermott et al., 2011*). SPT-TIRF-M analysis of JKCD4$^+$X4$^-$ cells transiently transfected with CXCR4$^{R334X}$-AcGFP (*Figure 4—videos 1–3*) demonstrated that, unlike the effect described for CXCL12, the VLPs containing gp120 triggered CXCR4$^{R334X}$ oligomerization (basal MSI 669 a.u.; Env(-) VLPs MSI 909 a.u.; gp120-VLPs MSI 1,730 a.u.; *Figure 4A*) and promoted a significant reduction in overall receptor diffusivity (basal, median $D_{1-4}$ = 0.021 µm$^2$s$^{-1}$; gp120-VLPs, median $D_{1-4}$ = 0.014 µm$^2$s$^{-1}$; *Figure 4B*) without significant differences in the percentage of trajectories with distinct type of diffusion, again most of them exhibited confined movement (*Figure 4—figure supplement 1*). Furthermore, gp120-VLP binding reduced the percentage of CXCR4$^{R334X}$ monomers and dimers (steady-state ~99%; gp120-VLPs ~80%), while concurrently increasing the percentage of nanoclusters containing ≥3 receptors/particle (basal ~1%; gp120-VLPs ~20%; *Figure 4C*).

All these data and other previously reported findings (*García-Cuesta et al., 2022*), indicate that CXCL12 triggers CXCR4 clustering at the cell membrane but does not induce CXCR4$^{R334X}$ oligomers. By contrast, gp120-VLPs binding stabilized CD4 complexes with both CXCR4 and CXCR4$^{R334X}$ and promoted oligomerization of both co-receptors. It is thus plausible that the conformation of CXCR4 and CXCR4$^{R334X}$ may differ between both experimental conditions.

## CD4 expression alters the conformation adopted by CXCR4

Our results support a model where CXCL12 binds to either CXCR4 or CXCR4$^{R334X}$ (*García-Cuesta et al., 2022*). By contrast, X4-gp120, whether alone or within the viral context, associates these co-receptors when they are complexed with CD4. It is known that CD4/CXCR4 complexes might facilitate co-operative interactions with HIV-1 during viral adsorption and/or entry into human leukocytes (*Singer et al., 2001*).

To confirm the interaction between CD4 and CXCR4 (*Martínez-Muñoz et al., 2014*) and to assess whether CXCR4$^{R334X}$ could also interact with CD4, we conducted FRET analyses. Our results demonstrated positive FRET signals for both complexes: CD4/CXCR4 (FRET$_{50}$=2.71) and CD4/CXCR4$^{R334X}$ (FRET$_{50}$=0.40; *Figure 5A and B*). As a control, we observed minimal residual energy transfer in cells co-transfected with CD4-CFP and 5HT$_{2B}$-YFP (*Figure 5C*). The presence of CD4/CXCR4 and CD4/CXCR4$^{R334X}$ heterodimers was also investigated in silico using alphaFold3 (*Abramson et al., 2024*; *Figure 5—figure supplement 1*). We focused on the interaction between the transmembrane and intracellular fragments of CD4 and both CXCR4 and CXCR4$^{R334X}$. The predicted template modeling (pTM) scores for both complexes were close to 0.7, suggesting a reliable prediction of these

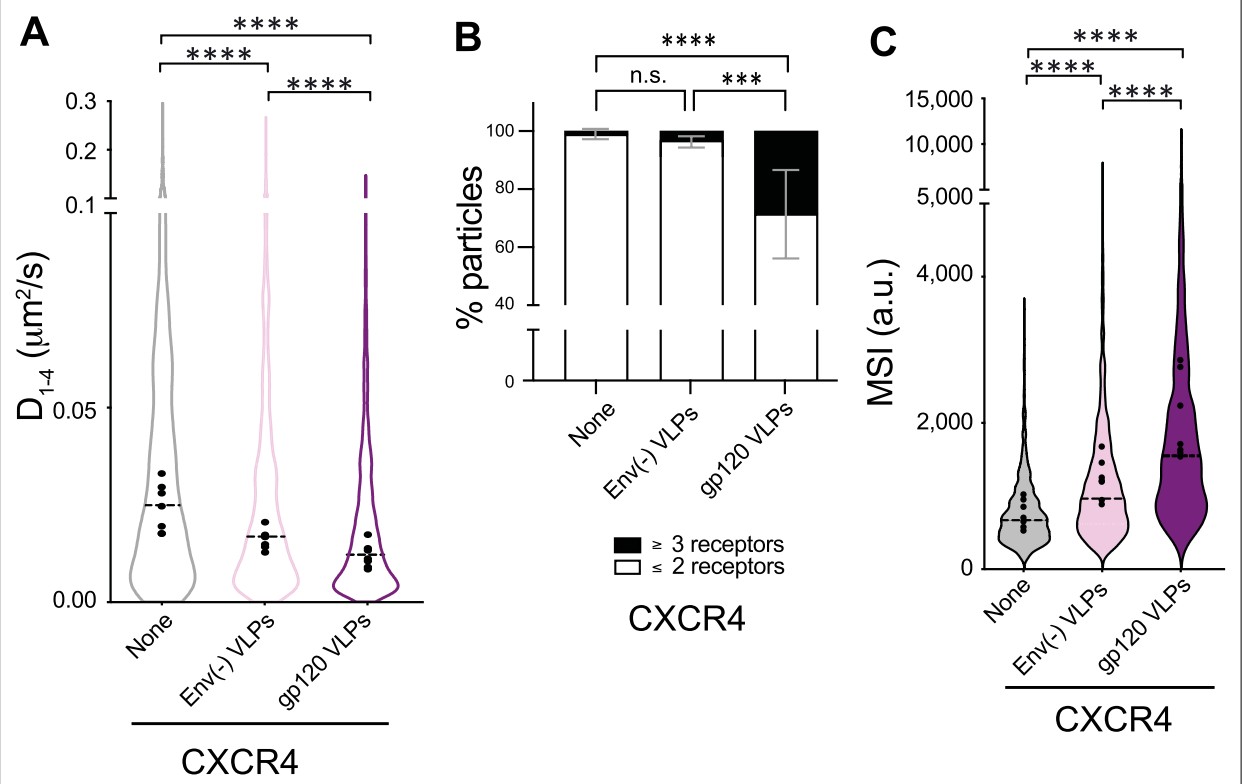

**Figure 3.** gp120 VLPs modulate CXCR4 dynamics and nanoclustering. Single-particle tracking analysis of JKCD4⁺X4⁻ cells transiently transfected with CXCR4-AcGFP, on fibronectin (FN)-, FN +VLPs-, or FN +gp120 VLPs-coated coverslips (1087 particles in 159 cells on FN; 1400 in 153 cells on FN +VLPs and 1061 in 160 cells on FN +gp120 VLPs) n=6. (**A**) Diffusion coefficients ($D_{1-4}$) of mobile particles at the membrane of cells treated as indicated. Figure shows the mean value of each experiment (black circles) and the median of all trajectories (dotted black lines; n=6; ****p≤0.0001). (**B**) Frequency of CXCR4-AcGFP particles containing monomers and dimers (≤2) or nanoclusters (≥3) in cells treated as indicated. Mean ± SD calculated from mean spot intensity (MSI) values of each particle as compared with the value of monomeric CD86-AcGFP (980±86 a.u., **p≤0.05, **p≤0.01, ****p≤0.0001). (**C**) Intensity distribution (arbitrary units, a.u.) from individual CXCR4-AcGFP trajectories on cells treated as indicated. Graph shows the distribution of all trajectories, with the mean value of each experiment (black circles) and the median of all trajectories (dotted black lines; n=6; ****p≤0.0001). Statistical significance was determined by non-parametric Kruskal-Wallis tests followed by Dunn's test for panels **A** and **C**, and by two-way ANOVA in panel **B**.

The online version of this article includes the following video and figure supplement(s) for figure 3:

**Figure supplement 1.** Classification of mobile receptor trajectories.

**Figure 3—video 1.** Representative video of CXCR4-AcGFP on live JKCD4+X4- cells treated with DMSO and captured by SPT-TIRF, showing the diffusion of CXCR4 particles (monomers, dimers, and nanoclusters) at steady state (FN).
https://elifesciences.org/articles/110354/figures#fig3video1

**Figure 3—video 2.** Representative video of CXCR4-AcGFP on live JKCD4+X4- cells treated with Env(-) VLPs and captured by SPT-TIRF, showing the diffusion of CXCR4 particles (monomers, dimers, and nanoclusters) at steady state (FN).
https://elifesciences.org/articles/110354/figures#fig3video2

**Figure 3—video 3.** Representative video of CXCR4-AcGFP on live JKCD4+X4- cells treated with gp120 VLPs and captured by SPT-TIRF, showing the diffusion of CXCR4 particles (monomers, dimers, and nanoclusters) at steady state (FN).
https://elifesciences.org/articles/110354/figures#fig3video3

interactions. The modeling also indicated a preferred association of CD4 with CXCR4 between transmembrane helices IV and V, although further analysis is needed to precisely determine the interaction site. These data align with previous reports demonstrating a constitutive association between CD4 and CXCR4 that can influence HIV-1 infection (*Martínez-Muñoz et al., 2014*; *Basmaciogullari et al., 2006*; *Zaitseva et al., 2005*; *Lapham et al., 1999*; *Lee et al., 2000*) and our results extend this observation to the naturally occurring CXCR4^R334X mutant. Furthermore, we observed increased FRET efficiency in both CD4-CXCR4 and CD4-CXCR4^R334X complexes upon gp120-VLPs binding (*Figure 5D and E*), confirming that the VLPs induce conformational changes in these heterodimers.

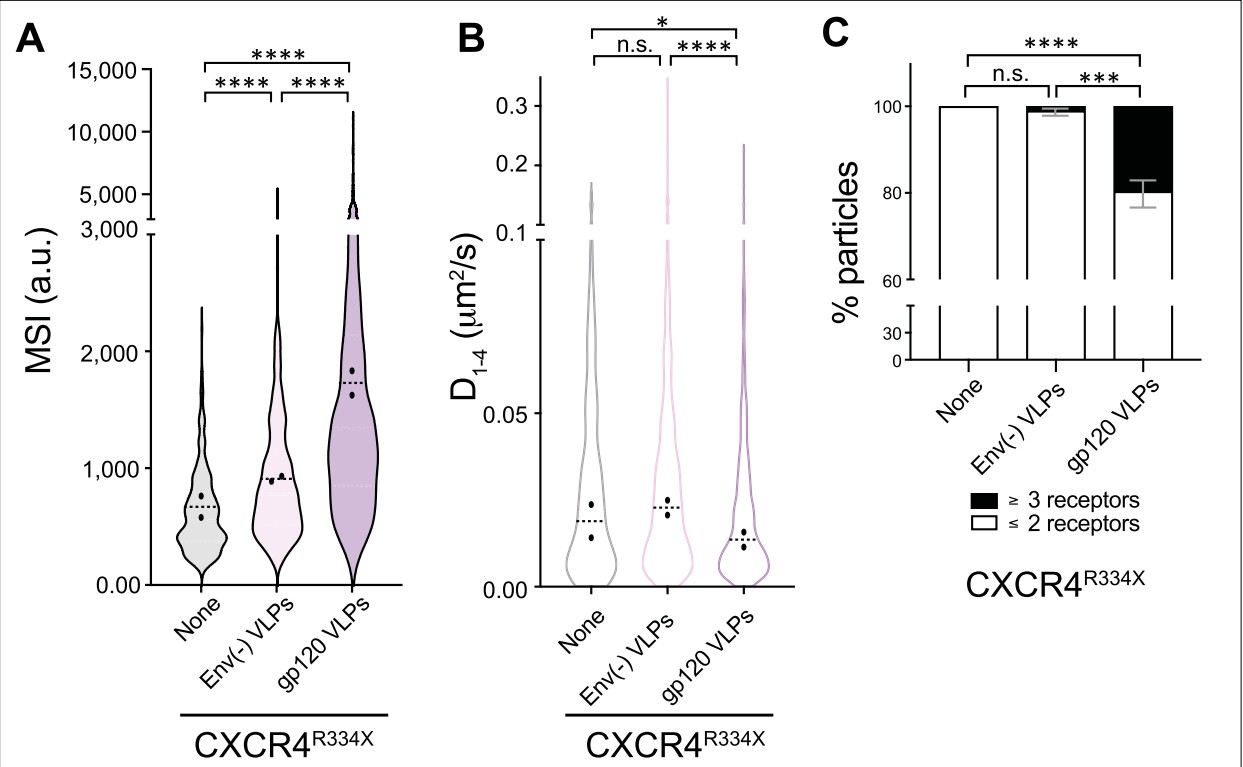

**Figure 4.** gp120 VLPs modulate CXCR4[R334X] dynamics and nanoclustering. Single-particle tracking analysis of JKCD4+X4- cells transiently transfected with CXCR4[R334X]-AcGFP, on fibronectin (FN)-, FN +VLPs-, or FN +gp120 VLPs-coated coverslips (341 particles in 63 cells on FN; 610 in 54 cells on FN +VLPs and 707 in 63 cells on FN +gp120 VLPs) n=2. (**A**) Intensity distribution (arbitrary units, a.u.) from individual CXCR4[R334X]-AcGFP trajectories on cells treated as indicated. Graph shows the distribution of all trajectories, with the mean value of each experiment (black circles) and the median of all trajectories ± SD (dotted black lines; n=2; ****p≤0.0001). (**B**) Diffusion coefficients ($D_{1-4}$) of mobile single particle trajectories at the membrane of cells treated as indicated. Figure shows the mean value of each experiment (black circles) and the median of all trajectories (dotted black lines; n=2; n.s. not significant, *p≤0.05, ****p≤0.0001). (**C**) Frequency of CXCR4[R334X]-AcGFP particles containing monomers plus dimers (≤2) or nanoclusters (≥3), ± SD calculated from mean spot intensity values of each particle as compared with the value of monomeric CD86-AcGFP (*p≤0.05, ***p≤0.001). Statistical significance was determined by non-parametric Kruskal-Wallis tests followed by Dunn's test for panels A and C, and by two-way ANOVA in panel B.

The online version of this article includes the following video and figure supplement(s) for figure 4:

**Figure supplement 1.** Classification of mobile receptor trajectories.

**Figure 4—video 1.** Representative video of CXCR4R334X-AcGFP on live JKCD4+X4- cells treated with DMSO and captured by SPT-TIRF, showing the diffusion of CXCR4 particles (monomers, dimers, and nanoclusters) at steady state (FN).
https://elifesciences.org/articles/110354/figures#fig4video1

**Figure 4—video 2.** Representative video of CXCR4R334X-AcGFP on live JKCD4+X4- cells treated with Env(-) VLPs and captured by SPT-TIRF, showing the diffusion of CXCR4 particles (monomers, dimers, and nanoclusters) at steady state (FN).
https://elifesciences.org/articles/110354/figures#fig4video2

**Figure 4—video 3.** Representative video of CXCR4R334X-AcGFP on live JKCD4+X4- cells treated with gp120 VLPs and captured by SPT-TIRF, showing the diffusion of CXCR4 particles (monomers, dimers, and nanoclusters) at steady state (FN).
https://elifesciences.org/articles/110354/figures#fig4video3

Productive HIV-1 infection of CD4+ cells markedly diminishes cell-surface expression of CD4 (*Hoxie et al., 1986*; *Potash and Volsky, 1998*). However, the impact on chemokine receptors is less clear. While some studies report a complete loss of CCR5 surface staining on cells chronically infected with R5 viruses (*Chenine et al., 2000*), the effect on CXCR4 varies (*Chenine et al., 2000*; *Choi et al., 2008*). Using flow cytometry, we investigated how stimulation with X4-gp120 affects the cell-surface expression of CD4, CXCR4, and CXCR4[R334X]. Consistent with previous findings (*Balabanian et al., 2005*), CXCL12 induced rapid internalization of CXCR4 without altering CXCR4[R334X] levels at the cell surface. By contrast, X4-gp120 caused a gradual, slight decrease in the surface expression of both receptors.

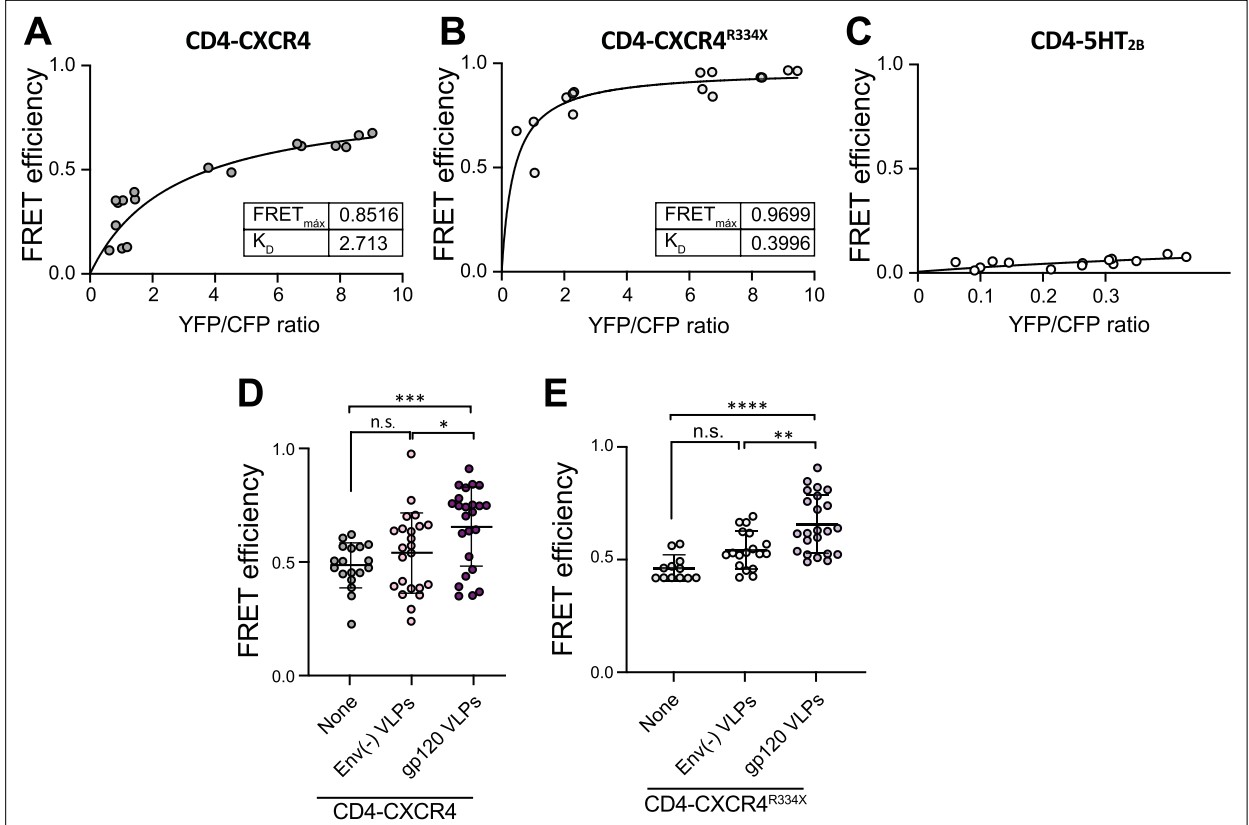

**Figure 5.** CD4 forms heterodimers with CXCR4 and CXCR4^R334X. FRET saturation curves generated using HEK-293T cells transiently transfected with a constant amount of CD4-CFP DNA (2 µg) and increasing amounts of (**A**) CXCR4-YFP (0.5–8.0 µg), (**B**) CXCR4^R334X-YFP (0.5–8.0 µg) or (**C**) 5HT$_{2B}$ DNA (0.5–12 µg). $K_D$ and FRET$_{max}$ values were calculated using a nonlinear regression equation for a single binding-site model (n=2). (**D**) FRET efficiency in HEK-293 cells transiently transfected with CXCR4-YFP/ CD4-CFP (ratio 15:9), in the absence or presence of gp120 VLPs or Env(-) VLPs. Data shows FRET efficiency (arbitrary units, a.u.; mean ± SD; n=3; n.s. not significant, *p≤0.05, ***p≤0.001). (**E**) FRET efficiency in HEK-293 cells transiently transfected with CXCR4^R334X-YFP/CD4-CFP (ratio 15:9), in the absence or presence of gp120-VLPs or Env(-) VLPs. Data shows FRET efficiency (a.u.; mean ± SD; n=3; n.s. not significant, *p≤0.05, ***p≤0.001, ****p≤0.0001). Statistical significance was determined by unpaired t-test in panels **D** and **E**.

The online version of this article includes the following figure supplement(s) for figure 5:

**Figure supplement 1.** Predicted interaction between CD4 and CXCR4 or CXCR4R334X by AlphaFold3.

Furthermore, CXCL12 did not affect CD4 surface levels, whereas X4-gp120 induced a gradual decline in CD4 expression (*Figure 6A and B*).

Taken together, our FRET and internalization experiments suggest that the effects of X4-gp120 on CXCR4 and CXCR4^R334X differ from those triggered by CXCL12 and are dependent on their interaction with CD4.

Next, we investigated whether CD4/CXCR4^R334X complexes could also support primary HIV-1 infection. Flow cytometry analysis showed no significant difference in the ability of Jurkat cells expressing either CXCR4 or CXCR4^R334X to bind X4-gp120 (*Figure 7A*). In agreement, in vitro fusion assays using cells expressing CD4/CXCR4 or CD4/CXCR4^R334X, and target cells expressing HIV pHXB2 Env, demonstrated a significant increase in fusion events and syncytium formation in both Jurkat cell types (*Figure 7B and C*). We then tested whether PBMCs isolated from a WHIM patient and healthy donors were equally susceptible to infection by a primary X4 HIV-1$_{NL4-3}$ viral strain. We first analyzed the expression of CD4, and CXCR4 or CXCR4^R334X on these PBMC samples by flow cytometry (*Figure 7— figure supplement 1*). Subsequently, cells were stimulated with PHA and IL-2, and 48 hoursr later, inoculated with a primary X4 HIV-1$_{NL4-3}$ virus at a MOI of 0.001 for 120 minutes. ELISA measurements of p24 levels in the culture medium at various time points revealed similar viral infection rates in both Healthy and WHIM PBMCs (*Figure 7D*).

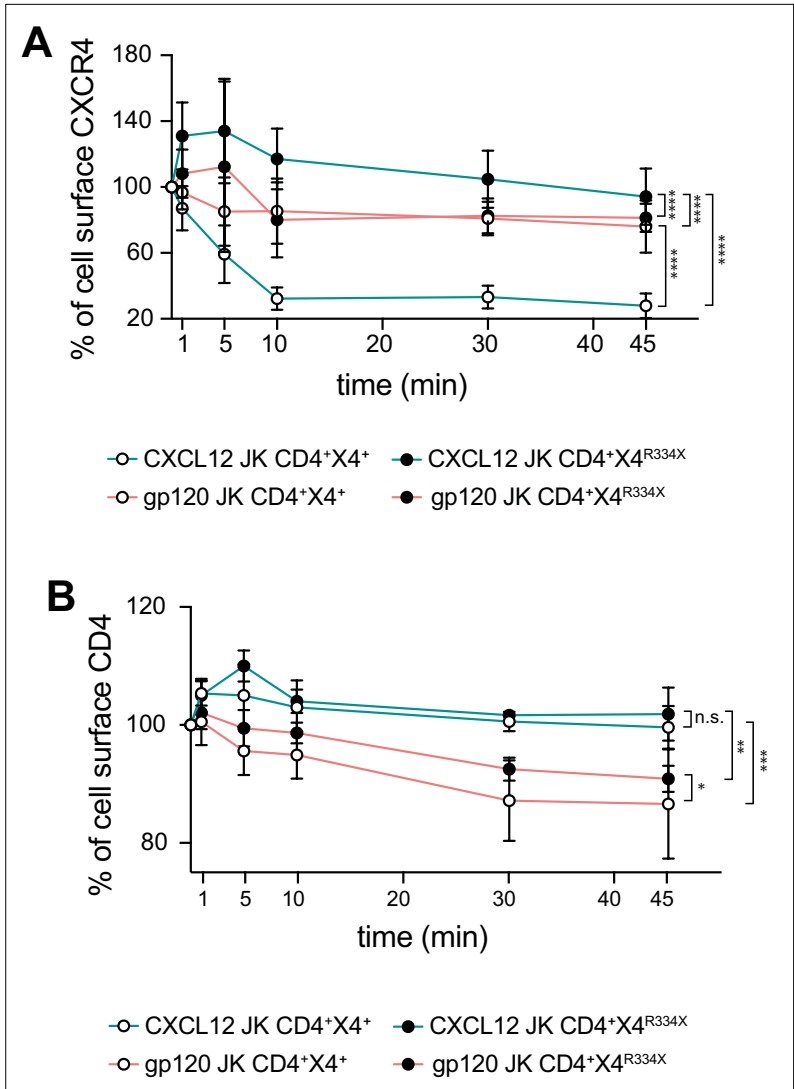

**Figure 6.** X4-gp120 promote similar internalization patterns of CXCR4 and CXCR4$^{R334X}$ receptors. (**A**) Surface receptor expression of CXCR4 (white dots) or CXCR4$^{R334X}$ (black dots), after stimulation with CXCL12 (blue lines) or X4-gp120 (red lines). Results show mean ± SD of the percentage of receptor expression at the cell surface (n=3). (**B**) Surface receptor expression of CD4 in JK CD4$^+$ CXCR4$^+$ (white dots) or JK CD4$^+$ CXCR4$^{R334X}$ (black dots), after stimulation with CXCL12 (blue lines) or X4-gp120 (red lines). Results show mean ± SD of the percentage of receptor expression at the cell surface (n=3). Statistical significance was determined by one-way-ANOVA of AUC (*p≤0.05, **p≤0.01).

Collectively, these data indicate that T cells from a WHIM patient exhibit similar infection and viral replication rates with those isolated from healthy donors. Furthermore, these data suggest that HIV-1 might modulate the conformation adopted by CXCR4 at the cell membrane, which is associated with HIV-1 infection. The CXCR4 conformation stabilized by HIV-1 binding likely differs from that induced by CXCL12 binding and therefore supports a direct effect of the interactions with CD4 in establishing a permissive conformation of CXCR4 for HIV-1 infection.

## Discussion

The process of HIV-1 infection begins with the binding of the trimeric HIV-1 Env glycoprotein to the CD4 receptor on the target cell surface (*Wyatt and Sodroski, 1998*; *Blumenthal et al., 2012*; *Chen, 2019*). Research indicates that when the viral and cellular membranes are spatially distant, the HIV-1 Env trimers initially engage only a single CD4 molecule, and as the Env moves closer to the membrane

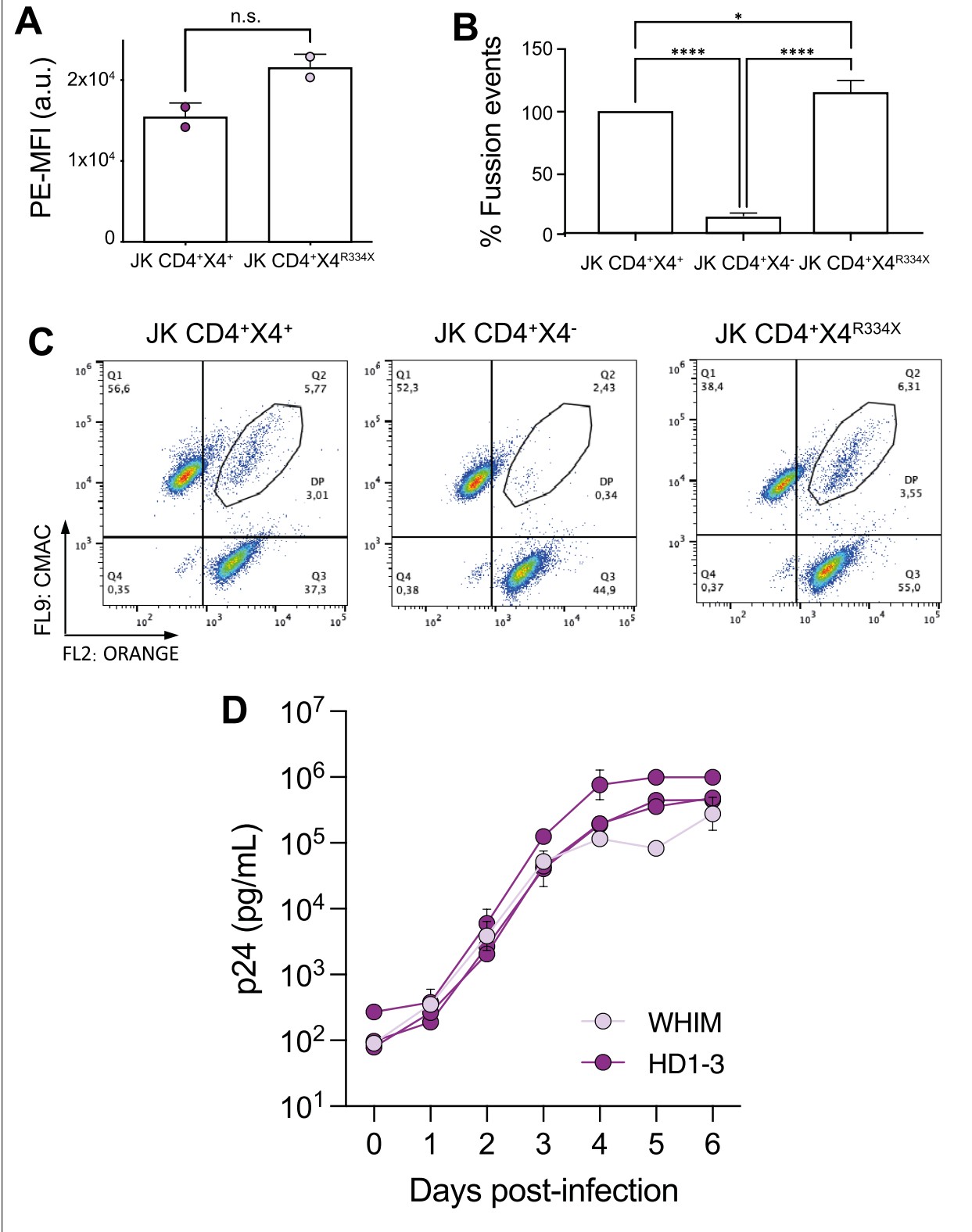

**Figure 7.** CD4/CXCR4 and CD4/CXCR4^R334X complexes support similar HIV-1 infection. The presence of CXCR4^R334X on JKCD4+ cells does not alter gp120 binding and increases fusion events with target cells expressing HIV pHXB2 envelope. (**A**) Binding of X4-gp120 to target cells expressing CD4 and CXCR4 or CD4 and CXCR4^R334X analyzed by flow cytometry. Cells were incubated with 0.3 mg/mL of X4-gp120 at 37 °C for 30 min. Data show MFI (arbitrary units, a.u.) mean ± SD; (n=2). Statistical significance was determined using Student's t-test (n.s.=not significant). (**B**) Cell-cell fusion between

*Figure 7 continued on next page*

*Figure 7 continued*

JKHXBc2-expressing HIV-1 envelope and different target cells (JKCD4+CXCR4+, JKCD4+CXCR4-, and JKCD4+CXCR4R334X). Prior to co-culture, each cell type was loaded with the corresponding cell-tracker. Data show the percentage of fusion events ± SD (n=6). We used as reference the fusions events detected in JKCD4+CXCR4+ cells (100%). Statistical significance was determined by one-way-ANOVA (*p<0.05, ****p≤0.0001). (C) Representative biparametric histograms from cells in B showing CMAC *versus* orange fluorophores. (D) Human PBMCs isolated from a WHIM patient (WHIM) and three healthy donors (HD1-3) in two independent experiments were infected with X4-pseudotyped HIV-1NL4-3 (MOI: 0.001). At 2 hr post infection (p.i.), supernatant samples were obtained at different time points (days post-infection) and p24 levels (pg/mL) in each sample were determined using a commercial ELISA. Results show mean ± SD (n=2).

The online version of this article includes the following figure supplement(s) for figure 7:

**Figure supplement 1.** Characterization of CD4+T cell populations in blood from healthy controls and a WHIM patient.

and adopts an open conformation, it gains the ability to bind a second and a third CD4 molecule (*Li et al., 2023*). CD4 binding thus induces critical conformational changes within the gp120 subunit of Env that enable subsequent engagement of the co-receptors CCR5 or CXCR4. Receptor and co-receptor engagement trigger further conformational changes in the Env gp41 subunits, ultimately mediating the necessary fusion of the viral and host cell membranes.

Competitive receptor inhibitors, such as soluble synthetic CD4 (sCD4), synthetic CD4 peptides, and anti-CD4 binding site antibodies, have been shown to effectively block infection both in vitro (*Traunecker et al., 1989*; *Nara et al., 1989*) and in vivo (*Schooley et al., 1990*; *Kahn et al., 1990*). Notably, virus-like nanoparticles displaying clustered membrane-associated CD4 demonstrated significantly greater efficacy at blocking infection compared with sCD4, CD4-Ig, or the broadly neutralizing monoclonal antibody 3BNC117 (*Hoffmann et al., 2020*). These findings collectively suggest that the virus may induce CD4 clustering to promote cell infection. Consistent with this idea, single-molecule super-resolution imaging has revealed that CD4 molecules on the cell membrane exist predominantly as individual molecules or small clusters (up to four molecules), and that the size and number of these clusters increase upon virus binding or gp120 activation (*Yuan et al., 2021*). These data therefore support a model where viral binding triggers a nanoscale reorganization of CD4 on the plasma membrane, a process necessary for cell infection. This observation aligns with substantial evidence indicating that various cell-surface receptors are organized into clusters (*Hartman and Groves, 2011*; *Schamel and Alarcón, 2013*). For instance, T-cell receptors on T cells coalesce into nanoclusters within and around immune synapses prior to signal transduction (*Razvag et al., 2019*; *Lillemeier et al., 2010*). Receptor clustering has been also demonstrated for numerous other transmembrane receptors, such as neurotransmitter receptors (*Verstreken and Bellen, 2002*) and immune receptors (*Germain, 1997*; *Cairo, 2007*). Furthermore, studies analyzing receptor dynamics during cell movement using SPT have shown that chemokine-mediated receptor clustering is essential for cells to accurately sense chemoattractant gradients (*Martínez-Muñoz et al., 2018*; *García-Cuesta et al., 2022*).

Here, using SPT-TIRF-M analysis on Jurkat cells, we found that both recombinant X4-gp120 and VLPs displaying X4 HIV-1 Env (with fewer gp120 molecules) actively promote CXCR4 clustering and modify its membrane dynamics in a CD4-dependent manner. Our data show that, similar to the effect of CXCL12, gp120 binding to CXCR4 leads to a significant decrease in the number of monomers and dimers, while increasing the proportion of larger, generally immobile nanoclusters. Previous studies utilizing immunoelectron microscopy have suggested that CCR5, CXCR4, and CD4 are mainly localized on microvilli and tend to form distinct, uniform microclusters in all cell types examined (*Singer et al., 2001*). This clustering is thought to enhance co-operative interactions with HIV-1 during virus adsorption and subsequent entry into human leukocytes (*Singer et al., 2001*). Single-molecule force spectroscopy has also revealed that the force and duration required to break apart the gp120/CCR5/CD4 complex are greater than those required for the gp120/CD4 bond (*Chang et al., 2005*). The formation of CCR5/CD4/CXCR4 oligomers is implicated in reducing the infectivity of X4 HIV-1 in cells that also express CCR5 (*Martínez-Muñoz et al., 2014*).

The data suggest that exposure to HIV-1 triggers several rearrangements of cell receptors, highlighting the significance of these changes in facilitating viral entry into target cells. Receptor clustering, a process known to enhance cell sensitivity (*Bray, 1995*), also promotes efficient cell signal transmission (*Cho and Stahelin, 2005*), and increases the robustness of signaling systems (*Gurry et al., 2009*). Indeed, it is well established that HIV-1 promotes CD4- and CXCR4-mediated signaling events that facilitate viral entry and infection of host cells (*Balabanian et al., 2004*; *Popik et al., 1998*;

*Davis et al., 1997*; *Pasquereau and Herbein, 2022*; *François and Klotman, 2003*). We observed that gp120 triggered the phosphorylation of Lck, Akt, and ERK1/2, and promoted cell polarization, although this latter effect was less pronounced than that induced by CXCL12. Furthermore, in lipid bilayer assays using ICAM-1 as a substrate, gp120-mediated Jurkat cell adhesion but did not significantly promote cell migration compared with CXCL12. While soluble gp120 has been used to investigate certain HIV-1 effects on cells (*Balabanian et al., 2004*; *Yoder et al., 2008*; *Deng et al., 2016*), employing saturating concentrations of the glycoprotein might lead to non-specific effects on the dynamics of receptors and co-receptors at the cell membrane. Moreover, the conformation of recombinant gp120 may not accurately reflect its physiological structure on intact HIV-1 particles. Within virions, the Env glycoprotein forms heterotrimeric gp120 non-covalently associated with three gp41 molecules. To address these limitations, we generated VLPs displaying the X4 HIV-1 Env. Super-resolution microscopy confirmed the maturity of the gp120-VLPs, expressing a very low number of gp120 trimers on their surface, even fewer than the 7–14 Env trimers per virus particle previously reported on primary isolated virions (*Zhu et al., 2003*; *Chertova et al., 2002*; *DeSantis et al., 2016*). Similar to soluble recombinant gp120, these VLPs also induced CXCR4 clustering and altered receptor dynamics. These findings thus confirm that the effects observed with X4-gp120 are not artifacts resulting from the conformation of the soluble gp120 or to the use of saturating concentrations. Receptor clustering was initiated through binding of gp120-VLPs to CD4 and CXCR4, as neither X4-gp120 nor the VLPs associated with CXCR4 in the absence of CD4. We confirmed this using a mutant CXCR4, CXCR4$^{R334X}$, which does not oligomerize in the presence of CXCL12 (*García-Cuesta et al., 2022*). CXCR4$^{R334X}$ is found in cells of patients with WHIM syndrome, a rare combined immunodeficiency characterized by the presence of warts, hypogammaglobulinemia, recurrent bacterial infections, and myelokathexis symptoms (*Liu et al., 2012*). In these patients, the inability of CXCL12 to induce receptor oligomerization leads to defects in actin dynamics, preventing proper sensing of chemoattractant gradients (*García-Cuesta et al., 2022*; *García-Cuesta et al., 2024*). We investigated how VLPs containing the X4 HIV-1 Env affected the dynamics of CXCR4$^{R334X}$ in cells expressing CD4. Surprisingly, the X4 HIV-1 Env caused a significant reduction in CXCR4$^{R334X}$ monomers and dimers, while increasing the proportion of larger nanoclusters, which were generally immobile. These findings suggest that CXCL12 and gp120 VLPs have different effects on chemokine receptor dynamics, leading us to hypothesize that the structure of CXCR4, whether or not it is associated with CD4, could explain these observations. AlphaFold predictions and FRET analysis confirmed the formation of CD4/CXCR4 complexes (*Martínez-Muñoz et al., 2014*) and supported the existence of CD4/CXCR4$^{R334X}$ heterodimers. In both cases, gp120-VLPs binding altered the conformations. These findings correlate with the syncytia formation observed in in vitro fusion experiments between cells expressing the X4 HIV-1 Env and target cells expressing either CD4/CXCR4 or CD4/CXCR4$^{R334X}$, and with the infection of PBMCs from both WHIM patients and healthy donors when incubated with primary X4-HIV-1. These findings also indicate that WHIM mutations do not protect against HIV-1 infection, consistent with a previous in vitro study showing that CD4$^+$U87 cells expressing CXCR4$^{R334X}$ or CXCR4 are equally susceptible to a luciferase-expressing pseudotyped virus infection (*Hernandez et al., 2003*). To better reflect natural infection kinetics, we employed fully replication-competent viruses rather than pseudotyped systems. Furthermore, we monitored viral dynamics over a seven-day period, providing a longitudinal perspective that extends beyond the limitations of a standard 24 hr snapshot. Beyond inducing CXCR4 aggregation, we observed that X4-gp120 promoted the internalization of CD4 and CXCR4 and, surprisingly, also triggered the internalization of CXCR4$^{R334X}$. This mutant receptor cannot internalize in response to CXCL12 because it lacks the last 19 C-terminal amino acids, which are necessary for GRK-mediated Ser/Thr phosphorylation and β-arrestin recruitment (*Liu et al., 2012*; *Kumar et al., 2023*). Some studies suggest that HIV-1 glycoproteins can reduce CD4 and CXCR4 levels during HIV-1 entry (*Choi et al., 2008*; *Geleziunas et al., 1994*; *Hubert et al., 1995*), proposing receptor-mediated endocytosis as an alternative HIV-1 entry mechanism (*Daecke et al., 2005*; *Aggarwal et al., 2017*; *de la Vega et al., 2011*; *Miyauchi et al., 2009*; *Carter et al., 2011a*; *van Wilgenburg et al., 2014*). Other reports even indicate a ligand-mediated co-endocytosis of CD4 and the chemokine receptors during HIV-1 entry (*Toyoda et al., 2015*; *Venzke et al., 2006*; *Gobeil et al., 2013*). HIV-1-mediated endocytosis might also explain the reduction of CD4 and CXCR4$^{R334X}$ at the cell membrane and the similar infection rates in PBMCs from WHIM patients and healthy donors. Although direct evidence for the internalization of CD4 and CXCR4 as complexes is lacking, their co-localization in lipid rafts during HIV-1 infection

(*Mañes et al., 2000*; *Popik et al., 2002*; *Neel et al., 2005*) and their ability to form heterocomplexes (*Martínez-Muñoz et al., 2014*) strongly suggest they could be endocytosed together. Therefore, we hypothesize that gp120 binding to CD4 stabilizes CD4/CXCR4 complexes, and that the conformation of CXCR4 within these complexes might differ from that of CXCR4 homodimers.

We also detected a residual but significant effect of Env(-) VLPs on the dynamics of both CXCR4 and of CXCR4$^{R334X}$. This is likely not receptor-mediated, as these control VLPs lacked the X4 HIV-1 Env. While further experiments are needed, a potential interaction between the lipids and or cell adhesion molecules derived from the cells where the virions were produced, and those in the cell membrane could explain this observation. It is well established that the lipid composition of the cell membrane influences ligand-mediated chemokine receptor oligomerization and dynamics (*Gardeta et al., 2022*; *Hauser et al., 2016*). Additionally, glycosaminoglycans, such as heparin and heparan sulphate, might play a role in the attachment of virions to various cell types (*Harrop and Rider, 1998*; *Mondor et al., 1998*; *Saphire et al., 2001*). Although their effects on R5 strains are debated (*Moulard et al., 2000*; *Ugolini et al., 1999*; *Dobrowsky et al., 2008*), the concentrations of these glycosaminoglycans vary considerably between cells (*Rabenstein, 2002*). Besides, some viruses bind lectins at the cell membrane, for instance, some microdomains of C-type lectin DC-SIGN have been described as portals for HIV-1 entry (*Cambi et al., 2004*; *Wu and KewalRamani, 2006*).

Our data indicate that HIV-1 not only affects CD4 dynamics, which is well known, but also alters the spatial distribution and dynamics of CXCR4 in a manner distinct from the effects of its natural ligand, CXCL12. While HIV-1 binding involves CD4/CXCR4 complexes, CXCL12 binds exclusively to CXCR4, potentially leading to different effects on CXCR4 conformations. These findings might also explain why HIV-1 induced the endocytosis of both CD4/CXCR4 and CD4/CXCR4$^{R334X}$ complexes, whereas CXCL12 did not mediate the internalization of either CD4 or the CD4/CXCR4$^{R334X}$ complex. This suggests that the interaction of both co-receptors with CD4 may result in different conformations of CXCR4 and CXCR4$^{R334X}$ compared with their respective homodimers. The interaction motif between CD4 and CXCR4 should be considered a crucial target for disrupting complex dynamics at the cell membrane, potentially opening new avenues for anti-HIV-1 therapies.

## Materials and methods

### Cells and reagents

HEK-293T cells, Jurkat human leukemia cells (JKCD4$^-$X4$^+$), and Daudi cells were obtained from the American Type Culture Collection (CRL-3216, CRL-10915 and CCL-213, respectively; ATCC, Rockville, MD). HEK-293CD4 cells, Jurkat CD4$^+$ cells (JKCD4$^+$X4$^+$), and Jurkat cells expressing an X4-tropic HIV-1 Env (JKHXBC2) were kindly provided by Drs. G. del Real (Instituto Nacional de Investigación y Tecnología Agraria y Alimentaria, Madrid, Spain) and J. Alcamí (Centro Nacional de Microbiología, Instituto de Salud Carlos III, Madrid, Spain). Cell lines obtained from the American Type Culture Collection (ATCC) were authenticated by the supplier using short tandem repeat (STR) profiling prior to distribution. Cell lines received from external academic laboratories were used as provided by the original investigators. No further authentication was performed during the course of this study. All cell lines tested negative for mycoplasma contamination. Where indicated, Jurkat cells lacking endogenous *CXCR4* (JKCD4$^+$X4$^-$; *García-Cuesta et al., 2022*) were electroporated with plasmids expressing wild-type CXCR4-AcGFP or mutant CXCR4$^{R334X}$-AcGFP receptors (20 µg), as described (*García-Cuesta et al., 2022*). Stable Jurkat cells expressing CXCR4$^{R334X}$ (JKCD4$^+$CXCR4$^{R334X}$) were generated by electroporation with CXCR4$^{R334X}$ and antibiotic selection (*García-Cuesta et al., 2022*).

Human peripheral blood mononuclear cells (PBMCs) were isolated from the blood of a patient with WHIM syndrome (CXCR4$^{R334X}$) or from healthy donors, and when required from buffy coats of healthy donors, which were obtained from the Centro de Transfusiones (Comunidad Autónoma de Madrid, Spain) by centrifugation through Percoll density gradients (760×g, 45 min, room temperature [RT]). CD4$^+$ cells were purified by negative selection using Dynabeads (Invitrogen, Thermo Fisher Scientific, Waltham, MA) and activated in vitro for 1 week with 50 U/mL of IL-2 (Teceleukin; Roche, Nutley, NJ) and 5 µg/mL phytohemagglutinin (Roche, Basel, Switzerland) to generate T cell blasts (*Gardeta et al., 2022*). The study using blood from WHIM patients and healthy donors was approved by the Institutional Review Board of the 12 de Octubre Health Research Institute (N° CEIm: 24/248), and was

conducted according to the principles of the Declaration of Helsinki. Informed consent was obtained from all patients.

Recombinant gp120 protein, HXBc2, was obtained from MyBiosource (#MBS43404, MyBiosource Inc, San Diego, CA). The following antibodies were used: anti-human CXCR4 monoclonal antibody (mAb; clone 44717) and phycoerythrin-conjugated anti-human CXCR4 mAb (clone 12G5; both from R&D Systems, Minneapolis, MN); goat F(ab')2 anti-mouse IgG-PE (Southern Biotech, Birmingham, AL); anti-human CD4 mAb (clone OKT4; Biolegend, San Diego, CA); anti-histidine mAb (clone AD1.1.10; R&D Systems); rabbit anti-gp120$_{IIIb}$ Ab (*Mañes et al., 2000*); rabbit anti-Gag p24 HIV-1 mAb (R&D Systems); and anti-phospho-AKT mAb (S473; #4060), anti-phospho-ERK1,2 mAb (T202/Y204; #9191), and anti-phospho-Lck mAb (Y505; #2751; all from Cell Signaling Technology, Danvers, MA); anti-tubulin mAb conjugated with rhodamine (Bio-Rad, Hercules, CA); phalloidin-TRITC (#P1951, Sigma-Merck, St Louis, MO); anti-ICAM 3 mAb (clone HP2/19) kindly donated by Dr. Francisco Sánchez Madrid (Instituto Sanitario Hospital Universitario La Princesa); goat anti-mouse-AF488 Ab (Thermo Fisher Scientific); anti-human gp120 mAb Fab fragments (clone 2G12; Polymun Scientific, Vienna, Austria); anti-human IgG Fab fragments (Jackson ImmunoResearch, West Grove, PA) conjugated to Abberior STAR RED (Abberior GmbH, Gottingen, Germany), kindly donated by Dr. Jakub Chojnacki (Germans Trias i Pujol Research Institute (IGTP)); anti-p24 HIV-1 (clone 37G12; Polymun Scientific) conjugated with Abberior STAR ORANGE. Human CXCL12 was obtained from PeproTech (Rocky Hill, NJ), and human CXCR4 was cloned into the pAcGFPm-N1 plasmid (Clontech Laboratories, Palo Alto, CA), as described (*Martínez-Muñoz et al., 2018*).

Fab fragments for staining in stimulated emission depletion (STED) microscopy were generated from the respective IgGs using the Fab Micro Preparation 3 Kit (Pierce, Thermo Fisher Scientific). The quality of Fab preparations was determined by measuring the absorbance of the eluted fractions of each conjugated antibody (at 280 nm and at the wavelength of maximum absorption of the fluorochrome). Anti-human Fab fragments were coupled to Abberior STAR RED dye *via* NHS-ester chemistry according to the dye manufacturer's instructions.

CellTracker Orange CMTMR and Blue CMAC (#C2927 and #C2110, respectively) were from Thermo Fisher Scientific.

## Gene constructs

Genes of HIV-1 gp120 (residues 31–507) from the isolate HXB2 (HIV-1$_{IIIB}$), with a C-terminal 6×histidine tag, were amplified by PCR from pHXB2-env (#1069 NIH-AIDS Reagent Program) using the oligonucleotides 5'NheI (5' TAACCGGTGCCAC CATGGACAGAGCCAAGCTGCTGCTGTTGCTGCTGCT GCTGCTGCTGCCTCAGGCTCAGGCCACTGAGAAGCTGTGGGTG 3') and 3'NotI (5' ATGCGGCC GCTCA GTGATGGTGATGGTGATGGGATCCACGCGGAACCAGCTGCACCACTCTTCT 3'), and were cloned into pIRES-PURO3 (#631619 from Clontech Laboratories).

The CD4 extracellular domain (residues 1–388), with a C-terminal 6×histidine tag, was amplified by PCR from CD4-pcDNA-3.1, kindly donated by Dr. Santos Mañes (Centro Nacional de Biotecnología, CSIC) (*Del Real et al., 2002*) using the oligonucleotides 5'AgeI (5' TAACCGGTATGAACCGGGGA GTCCCT 3') and 3' NotI (5' ATGCGGCCGCCTAGTATG GTGATGGTGATGCAAGTCCTCTTCAGAA ATGAGCTTTTGCTCGGGCAGAACCT TGAT 3'), and was cloned into pIRES-PURO3 (#631619 from Clontech Laboratories).

Stably-transfected HEK-293T cell lines were generated for each construct. Briefly, $0.5×10^6$ cells were seeded in DMEM supplemented with 10% FCS in a six-well plate 24 hr before transfection. Cells were then transfected with Expifectamine in OptiMem media (ExpiFectamine 293 transfection kit; #A14525 from Thermo Fisher Scientific). At 48 hr post-transfection, cells were transferred to DMEM containing 10% FBS and 2 μg/mL puromycin for selection ($1×10^4$ cells per well in a 96-well plate). Protein expression was confirmed by western blotting (using anti-gp120IIIB or anti-histidine mAbs, depending on the construct). Positive clones were expanded, frozen, and stored in liquid nitrogen.

## Purification of recombinant proteins: soluble HIV-1 gp120 and soluble CD4

HEK-293T cells expressing C-terminal his-tagged HXB2-gp120 or the extracellular CD4 domain were grown in DMEM containing 10% FBS and 2 μg/mL puromycin in 150 mm plates. The filtered supernatants were passed through a Nickel Agarose Extrachel column, Ni-NTA, (ABT technologies, Madrid,

Spain) at a flow rate of 0.5 mL/min. Each recombinant histidine-tagged protein was eluted utilizing a step-gradient protocol using a Tris 50 mM pH 8, NaCl 500 mM buffer containing 500 mM imidazole. The elution involved initial steps of 10% and 20% imidazole, followed by a linear gradient to 100% to ensure complete protein recovery. The eluted fractions were analyzed by SDS-PAGE gel electrophoresis. Those fractions containing the protein of interest were pooled and concentrated to a final volume of 0.5 mL. The concentrated sample was subjected to gel filtration chromatography using a S200 Increase (10/300) column (Cytiva, Freiburg, Germany). The eluted fractions were analyzed by SDS-PAGE and fractions containing the expected molecular size were pooled, aliquoted, and stored at –80 °C.

## Production of viral-like particles and lentiviral particles

Production of VLPs in HEK-293T cells included transfection with a plasmid encoding the X4-tropic HIV-1 Env (10 µg pHXB2env; #1069 NIH-AIDS Reagent Program) and a plasmid encoding a second-generation packaging system (7 µg psPAX2; #12260 Addgene, Watertown, MA). Supernatants were collected 48 hr post-transfection, filtered through 0.45 µm filters, and clarified by ultracentrifugation (247,000 × $g$, 2 hr, 4 °C) on a 20% sucrose cushion, using a Beckman SW55 rotor. The resulting VLPs were aliquoted and stored at –80 °C. Expression of gp120 and p24 was assessed by western blotting with specific antibodies. Each VLP batch was quantified using a p24 Quantikine ELISA kit (R&D Systems).

Production of lentiviral particles (LVPs) was performed as above with the co-transfection of a reporter gene (8 µg pIGFP; #PS100065, OriGene, Rockville, MD) instead of the double transfection. Supernatants were collected 48 hr post-transfection, filtered through 0.45 µm filters, aliquoted, and stored at –80 °C. Expression of gp120 and p24 was assessed by western blotting with specific antibodies. Each LVP batch was characterized in a transduction assay using LVPs transfected with the vesicular stomatitis virus G glycoprotein (VSVG; pCMV-VSV-G; #8454, Addgene) as a positive control.

## Western blotting

Cells (3×10⁶) were activated with CXCL12 (50 nM) or recombinant X4-gp120 (0.3 µg/mL) at the time points indicated and then lysed in RIPA detergent buffer containing 1 mM PMSF, 10 µg/mL aprotinin, 10 µg/mL leupeptin, and 10 µM sodium orthovanadate (30 min, 4 °C). Cell extracts were analyzed by western blotting using specific antibodies. Densitometric evaluation of western blots was performed using ImageJ software (NIH, Bethesda, MD).

## Flow cytometry

Cells (2×10⁵/well) were incubated with specific antibodies (30 min, 4 °C), and mean fluorescence intensity (MFI) was determined on a Gallios or FC500 flow cytometer (Beckman Coulter). When required, Jurkat cells expressing CD4 (JKCD4+X4+ and JKCD4+X4-) were incubated with X4-gp120-His (0.3 µg/mL), followed by staining with an anti-histidine-PE mAb. Similarly, Daudi cells were incubated with a mixture of X4-gp120 (0.3 µg/mL) and soluble hCD4-his (0.2 µg/mL), and subsequently stained with an anti-histidine-PE mAb.

For human PBMCs, 100 µL of whole blood from healthy controls and a WHIM patient were analyzed by flow cytometry for the expression of CD3-APC (IM2467), CD19-PC5.5 (A66328), CD4-FITC (A07750), CD8-PC7 (737661; all from Beckman Coulter Inc, Brea, CA), and CXCR4-PE (306506, Biolegend) using a Gallios flow cytometer and FlowJo software.

Receptor internalization was evaluated after cell activation (5×10⁵ cells/well) with 50 nM CXCL12 or 0.3 µg/mL of recombinant X4-gp120 at the indicated time points. Cells were incubated with an anti-CXCR4 mAb (clone 44717, 30 min, 4 °C), followed by a PE-coupled goat anti-mouse IgG (30 min, 4 °C) or, when required, with anti-CD4 (clone OKT4, 20 min, 4 °C), and analyzed in a Cytoflex cytometer (Beckman Coulter). Results are expressed as a percentage of mean spot intensity (MSI) of treated cells relative to that of unstimulated cells.

To evaluate VLPs by flow cytometry, particles were coupled to latex beads (4 mm, 4% w/v Aldehyde/Sulfate latex; Invitrogen, Eugene, OR). After sonication (5 min, RT), beads were mixed with VLPs at a ratio of 1:1 v/v (15 min, RT) in 1% casein-PBS solution (Bio-Rad). Reactive groups were blocked with 100 mM glycine (60 min, 4 °C, with continuous rocking). Beads coupled to VLPs were washed twice by centrifugation (3 min, 2000 × $g$) in washing buffer (PBS/BSA 0.5%), resuspended,

and incubated with the corresponding dilution of the recombinant soluble CD4 in casein-PBS solution (30 min, RT). VLP-beads conjugates were washed three times (3 min, 2000 × *g*) with PBS staining buffer (PBS supplemented with 2% FBS, 1% BSA, and 0.2% sodium azide) and stained as above for flow cytometry.

## Lentiviral particle transduction assays

HEK-293 CD4 cells ($1.2 \times 10^4$ cells per well in a 96-well plate) were inoculated with serial dilutions of LVP-containing supernatants. Stock solutions were diluted in DMEM, 10% FCS, 1 mM pyruvate, and 2 mM glutamine. Sixteen hours later, the viral inoculum was replaced with fresh medium, and the cells were further incubated (48 hr, 37 °C). Viral transduction efficiency was determined by GFP fluorescence analysis. The medium was discarded, and cells were washed with PBS and fixed using 4% formaldehyde in PBS (20 min, RT). Fluorescence was imaged in a Tecan Spark Cyto plate reader (Tecan Group Ltd., Männedorf, Switzerland) after extensive washes with PBS. Images of 2456 × 2052 pixels at a 16-bit gray scale were acquired with a 4× objective. All images were captured using identical exposure settings (200 ms), except for wells where cells were transduced with LVPs expressing VSVG, for which an exposure time of 80ms was used. Mean fluorescence intensity of GFP signal of each image was quantitated using Fiji/ImageJ v2.3.0/1.53t software.

## Cell-cell fusion assay

The JKHXBC2 cell line was co-cultured with the indicated Jurkat target cells at a 1:1 ratio in 96-well flat-bottom plates (16 hr, 37 °C). Prior to co-culture, each cell type was stained with vital probes Cell-Tracker Orange CMTMR and Blue CMAC, respectively. Double-stained events were subsequently analyzed using a Gallios Analyzer cytometer (Beckman Coulter). Results are shown as the percentage of fusion events ± SD, using as a reference the fusions events detected in JKCD4⁺CXCR4⁺ cells.

## Cell adhesion/migration on planar lipid bilayers

Planar lipid bilayers were prepared as reported (*Carrasco et al., 2004*). Briefly, unlabeled GPI-linked intercellular adhesion molecule 1 (ICAM-1) liposomes were mixed with 1,2-dioleoyl-phosphatidyl choline. Membranes were assembled in FCS2 chambers (Bioptechs, Butler, PA), blocked with PBS containing 2% FCS (60 min, RT), and coated with CXCL12 (200 nM, 30 min, RT) or gp120 (0.3 µg/ml, 30 min, RT). Cells ($3 \times 10^6$ CD4⁺ T blasts/mL) in PBS containing 0.5% FCS, 0.5 g/L D-glucose, 2 mM $MgCl_2$, and 0.5 mM $CaCl_2$ were then injected into the pre-warmed chamber (37 °C). Confocal fluorescence, differential interference contrast (DIC), and interference reflection microscopy (IRM) images were acquired on a Zeiss Axiovert LSM 510-META inverted microscope with a 40×oil-immersion objective. Imaris 7.0 software (Bitplane, Zurich, Switzerland) and ImageJ 1.49 v were used for qualitative and quantitative analysis of cell dynamics parameters, fluorescence, and IRM signals. The fluorescence signal of the planar bilayer in each case was established as the background fluorescence intensity. The frequency of adhesion (IRM⁺ cells) per image field was estimated as [n° of cells showing IRM contact/ total number of cells (estimated by DIC)]×100; similarly, we calculated the frequency of migration (cells showing an IRM contract and moving over time).

## STED assays

Purified particles (~1 µg of p24) were adhered to glass cover slips previously coated with 0.01% poly-L-lysine (Sigma) for 20 minutes. Cover slips were briefly fixed with 3% PFA/PBS and blocked using 2% BSA (Sigma)/PBS for 30 minutes. Particles were stained for Env using 10 ng/µL 2G12 Fab fragments and anti-human Abberior STAR RED-conjugated Fab fragments. Following immunostaining, particles were washed in permeabilization buffer (0.1% saponin, 0.5% BSA in PBS) and stained for Gag using 20 ng/µL 37G12 Ab conjugated with STAR ORANGE. The samples were briefly fixed using 3% PFA/ PBS again and were overlaid with SlowFade Diamond (Thermo Fisher Scientific). All steps were carried out at RT.

The VLPs were imaged using an Olympus IX83 inverted confocal microscope equipped with the Abberior STEDYCON STED system using a 60×/1.42NA objective. The following parameters were used during STED image acquisitions: pinhole size: 1 Airy; dwell time: 300 µs/pixel; field of view: 20 µm × 20 µm; and pixel size: 20 nm.

The mature state of VLPs, the percentage which express gp120 on their surface, and the intensity of the signal of gp120 per VLP, were quantified manually using ImageJ.

## Imaging of viral-like particles by transmission electron microscopy

The integrity of VLPs was examined by negative-stain electron microscopy on carbon grids. Samples were incubated on the grids and were treated with 2% uranyl acetate (30 s, RT). Grids were examined using a transmission electron microscope (1200-EX II; Jeol, Tokyo, Japan) at 100 kV, equipped with a Gatan Oneview CMOS camera.

## FRET saturation curves by sensitized emission

FRET analyses were performed as described (*Martínez-Muñoz et al., 2009*). Briefly, HEK-293T cells ($3.5 \times 10^5$ cells/well) were transiently transfected with cDNA encoding the fusion proteins using the poly-ethylenimine method (Sigma-Aldrich). For CD4/CXCR4 or CD4/CXCR4$^{R334X}$ heterodimers, the cells were co-transfected with a fixed amount of CD4-CFP (2 μg) and increasing amounts of CXCR4-YFP or CXCR4$^{R334X}$-YFP (0.5–8.0 μg). As a control, we used a fixed amount of CD4-CFP (2 μg) and increasing amounts of 5HT$_{2B}$-YFP (0.5–12 μg). We incubated cells with cDNA and poly-ethylenimine (5.47 mM in nitrogen residues) and 150 mM NaCl in serum-free medium, which was replaced after 4 hoursr by complete medium. At 48 hoursr post transfection, cells were washed twice in HBSS supplemented with 0.1% glucose and resuspended in the same solution. Total protein concentration was determined for whole cells using the Bradford Assay Kit (Bio-Rad). Cell suspensions (0.2 μg/μL) were added to black 96-well microplates and emission light was quantified using the Wallac Envision 2104 Multi-label Reader (Perkin-Elmer, Waltham, MA) as described (*Martínez-Muñoz et al., 2018*). To determine FRET$_{50}$ and FRET$_{max}$ values, curves were extrapolated from data using a nonlinear regression equation applied to a single binding site model with a 95% confidence interval (GraphPad PRISM software, San Diego, CA). When FRET at a fixed ratio was needed, HEK-293T cells were transiently co-transfected with a fixed ratio of CXCR4-YFP/CD4-CFP (15 μg and 9 μg, respectively) and CXCR4$^{R334X}$-YFP/CD4-CFP (9 μg and 9 μg, respectively in all cases). After 48 hoursr, the cells were treated with Env(-) VLPs (0.1 μg/mL of Gag p24) or gp120-VLPs (0.1 μg/mL of Gag p24) and FRET efficiency was evaluated (n=3, mean ± SD).

## Single-molecule TIRF imaging and analysis

Transfected cells expressing 8,500–22,000 receptors/cell (<4.5 particles/μm$^2$) were selected for detection and tracking analysis. Experiments were performed at 37 °C with 5% CO$_2$ using a TIRF microscope (Leica AM TIRF inverted microscope; Leica Microsystems, Wetzlar, Germany). Image sequences of individual particles (500 frames) were then acquired at 49% laser power (488 nm diode laser) with a frame rate of 10 Hz (100 ms/frame). The penetration depth of the evanescent field was 90 nm. Particles were detected and tracked using the U-Track2 algorithm (*Jaqaman et al., 2008*) implemented in MATLAB, as described (*Martínez-Muñoz et al., 2018*). MSI, number of mobile and immobile particles, and diffusion coefficients (D$_{1-4}$) were calculated from the analysis of thousands of single trajectories over multiple cells (statistics provided in the respective figure captions) using described routines (*Martínez-Muñoz et al., 2018*). The receptor number along individual trajectories was determined as reported (*Dorsch et al., 2009*), using the intensity of the monomeric protein CD86-AcGFP as a reference (*Figure 1—figure supplement 5*). Values were confirmed using single-step photobleaching analysis (*Martínez-Muñoz et al., 2018*; *García-Cuesta et al., 2022*).

## HIV-1 infection in T CD4$^+$ cells of a WHIM patient

All donors were female and under the age of 30. PBMCs were obtained from sero-negative fresh blood of a WHIM patient and three healthy donor controls using SepMate tubes (STEMCELL Technologies, Vancouver, Canada). Isolated PBMCs were cultured in RPMI supplemented with 20% FBS (Gibco), 1% penicillin/streptomycin (Capricorn Scientific GmbH, Ebsdorfergrund, Germany), 3 μg/mL PHA (Sigma-Aldrich), and 10 U/mL of IL-2 (Novartis, Basel, Switzerland) for 48 hoursr at 37 °C. Activated PBMCs were inoculated in vitro at a multiplicity of infection (MOI) of 0.001 with the HIV-1 NL4-3 lab strain for 1 hour. After removal of the viral inoculum, cultures were split into two wells and maintained in RPMI supplemented with 20% FBS, 1% penicillin/streptomycin, and 10 U/mL of IL-2. Supernatants were collected at different time points and replaced with fresh supplemented media.

The infection rate was determined by quantification of viral p24 in the supernatants using the Alliance HIV-1 p24 Antigen Elisa Kit (Perkin Elmer).

## Statistical analyses

All results were analyzed with GraphPad Prism software version 9. Cell polarization assays, using immunofluorescence or planar lipid bilayers comparing different conditions, were analyzed to determine significant differences between means using one-way analysis of variance (ANOVA) followed by Tukey's multiple comparisons test. Data related to the percentage of mature VLPs and their surface expression of gp120 obtained by STED microscopy, and MFI results by flow cytometry, were also analyzed by one-way ANOVA and Tukey's multiple comparisons test. A two-tailed Mann-Whitney non-parametric test was used to analyze the MFI of gp120 per VLP. The Kruskal-Wallis test followed by Dunn's test was used to analyze MSI and diffusion coefficients ($D_{1-4}$) of single particles in TIRF experiments. We used contingency tables to compare two or more groups of categorical variables, such as the percentages of mobile or immobile particles, and these were compared using a Chi-square test with a two-tailed p-value. Statistical differences were reported as n.s.=not significant $p > 0.05$, $*p \leq 0.05$, $**p \leq 0.01$, $***p \leq 0.001$, and $****p \leq 0.0001$.

## Acknowledgements

This work was supported by grants from the Spanish Ministry of Science and Innovation (PID2020-114980RB-I00 and PID2023-146301OB-I00) to MM and (PID2022-140651NB-I00) to CS, and (PID2022-139271OB-I00 and CB21/13/00063) to JM-P. EG-C and BS were supported by the program Apoyos Centros de Excelencia SO of the Spanish Ministry of Science and Innovation (SEV-2017-0712). AQ-F and SRG ere included in the doctoral program of the Department of Molecular Biosciences and of Biology, respectively, Universidad Autónoma de Madrid, and are supported by the Fondo de Personal Investigador (FPI) program of the Spanish Ministry for Science and Innovation (PRE2018-083201 and PRE2019-087966, respectively). EAB is included in the doctoral program of Biomedicine, University of Barcelona, and is supported by the FPI program of the Spanish Ministry for Science and Innovation (PRE2022-000847). RA-B is supported by the Garantía Juvenil program of the Regional Government of Madrid, Spain (CAM20_CNB_AI_07). LIGG receives funding from Institute of Health, Carlos III co-funded by Fondos FEDER Project FIS PI21/01642. We also acknowledge the technical help of the Advance Light Microscopy Unit at the CNB/CSIC. JMP has received research funding, consultancy fees, and lecture sponsorships from and has served on advisory boards for various companies (AbiVax, AstraZeneca, MSD, Gilead Sciences, ViiV Healthcare, Johnson & Johnson) outside the scope of this article.

## Additional information

### Competing interests

Javier Martinez-Picado: has received research funding, consultancy fees, and lecture sponsorships from, and has served on advisory boards for various companies (AbiVax, AstraZeneca, MSD, Gilead Sciences, ViiV Healthcare, Johnson & Johnson) outside the scope of this article. The other authors declare that no competing interests exist.

### Funding

| Funder | Grant reference number | Author |
| --- | --- | --- |
| Spanish National Plan for Scientific and Technical Research and Innovation | PID2020-114980RB-I00 | Mario Mellado |
| Fundación para el Conocimiento Madri+d | CAM20_CNB_AI_07 | Rosa Ayala-Bueno |
| Federación Española de Enfermedades Raras | FIS PI21/01642 | Luis Ignacio González-Granado |

| Funder | Grant reference number | Author |
| --- | --- | --- |
| Spanish National Plan for Scientific and Technical Research and Innovation | PID2023-146301OB-I00 | Mario Mellado |
| Spanish National Plan for Scientific and Technical Research and Innovation | PID2022-140651NB-I00 | César A Santiago |
| Spanish National Plan for Scientific and Technical Research and Innovation | PID2022-139271OB-I00 | Javier Martinez-Picado |
| Centro de Investigación Biotecnológica en Red de Enfermedades Infecciosas | CB21/13/00063 | Javier Martinez-Picado |
| Centro Nacional de Biotecnología | SEV-2017-0712 | Eva M García-Cuesta |
| Spanish National Plan for Scientific and Technical Research and Innovation | PRE2018-083201 | Sofia R Gardeta |
| Spanish National Plan for Scientific and Technical Research and Innovation | PRE2019-087966 | Adriana Quijada-Freire |
| Spanish National Plan for Scientific and Technical Research and Innovation | PRE2022-000847 | Eva Armendariz-Burgoa |

The funders had no role in study design, data collection and interpretation, or the decision to submit the work for publication.

## Author contributions

Adriana Quijada-Freire, Formal analysis, Investigation, Methodology, Writing – original draft; César A Santiago, Blanca Soler Palacios, Rosa Ayala-Bueno, Sofia R Gardeta, Enara San Sebastian, Eva Armendariz-Burgoa, Maria Carmen Puertas, Ricardo Villares, Methodology; Eva M García-Cuesta, Software, Formal analysis, Supervision, Investigation, Methodology, Writing – review and editing; Urtzi Garaigorta, Supervision, Validation, Investigation, Methodology; Luis Ignacio González-Granado, Resources, Investigation, Methodology; Jose Miguel Rodriguez Frade, Conceptualization, Resources, Data curation, Formal analysis, Supervision, Validation, Investigation, Methodology, Writing – original draft, Project administration; Jakub Chojnacki, Supervision, Validation, Methodology; Javier Martinez-Picado, Supervision, Investigation, Methodology, Project administration; Mario Mellado, Conceptualization, Resources, Supervision, Funding acquisition, Visualization, Writing – original draft, Project administration

## Author ORCIDs

César A Santiago https://orcid.org/0000-0002-5149-1722
Eva M García-Cuesta https://orcid.org/0000-0003-2311-4353
Eva Armendariz-Burgoa https://orcid.org/0009-0003-7173-5004
Ricardo Villares https://orcid.org/0000-0001-7562-6700
Urtzi Garaigorta https://orcid.org/0000-0002-0683-5725
Javier Martinez-Picado https://orcid.org/0000-0002-4916-2129
Mario Mellado https://orcid.org/0000-0001-6325-1630

## Ethics

The study using blood from WHIM patients and healthy donors was approved by the Institutional Review Board of the 12 de Octubre Health Research Institute (N°; CEIm: 24/248), and was conducted according to the principles of the Declaration of Helsinki. Informed consent was obtained from all patients.

Reviewer #1 (Public review): https://doi.org/10.7554/eLife.110354.2.sa1
Reviewer #2 (Public review): https://doi.org/10.7554/eLife.110354.2.sa2

Reviewer #3 (Public review): https://doi.org/10.7554/eLife.110354.2.sa3
Reviewer #4 (Public review): https://doi.org/10.7554/eLife.110354.2.sa4
Author response https://doi.org/10.7554/eLife.110354.2.sa5

## Additional files

### Supplementary files
MDAR checklist

### Data availability
All data generated or analysed during this study are included in the manuscript, figures, figure supplements and source data files.

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
