## [Editor Report · eLife Assessment]

This study provides **valuable** insights into how HIV-1 Env modulates the nanoscale organization and dynamics of the CXCR4 co-receptor on T cells, using quantitative imaging and functional approaches, the authors present **convincing** evidence that gp120 engagement promotes CD4-dependent clustering and altered mobility of CXCR4, distinct from the effects of the natural ligand CXCL12. Some concerns were raised regarding the interpretation of the single-particle tracking analyses, and additional clarification or analysis may help strengthen the conclusions. The physiological relevance of the findings could be further enhanced by validation with infectious virus and by more clearly integrating the CXCR4R334X mutant observations into the central mechanistic narrative. The work will be of interest to researchers studying HIV entry and membrane receptor organization.

[Editors' note: this paper was reviewed by Review Commons.]

---

## [Referee Report · Reviewer #1 (Public review)]

Summary:

This article provides new insights into the organisational changes of the X4-tropic HIV-1 co-receptor CXCR4 upon binding of the viral receptor-binding protein X4-gp120, either in its soluble form or when displayed as Env on virus-like particles (VLPs). The study employs single-particle tracking total internal reflection fluorescence (SPT-TIRF) microscopy to quantify the dynamics and clustering of CXCR4 on CD4+ T cells. The data show that CXCR4 clusters in the presence of X4-gp120 and VLPs, a phenomenon that is also observed for the primary HIV-1 receptor CD4. The authors also show that a WHIM mutant of CXCR4 (CXCR4-R334X) that does not cluster in the presence of its natural ligand, CXCL12, clusters in the presence of X4-gp120 and VLPs.

Major strengths:

The data are well presented, discussed, and supported by solid evidence. Literature is cited appropriately.

Major weaknesses:

The authors have addressed my concerns in the revised manuscript.

Significance:

In summary, the work is presented in a clear fashion, and the main findings are properly highlighted. The paper will be of interest to the broader virology community as well as to researchers studying cell receptor clustering. The findings are not entirely surprising because it has been shown previously that the binding of Env to CD4 mediates CD4 clustering, which would also suggest clustering of the co-receptor. Nonetheless, the paper provides strong evidence that CXCR4 clusters and changes its dynamics in the presence of CD4 and X4-gp120. Moreover, the evidence that X4-gp120 clusters CXCR4-R334X is of high interest as it suggests a different binding mechanism for X4-gp120 from that of the natural ligand CXCL12, raising questions for further research.

---

## [Referee Report · Reviewer #2 (Public review)]

Summary:

The author investigates how the HIV-1 Env glycoprotein modulates the nanoscale organisation and dynamics of the CXCR4 co-receptor on CD4⁺ T cells. The author demonstrates that HIV-1 Env induces CXCR4 clustering distinct from that triggered by its natural ligand (CXCL12), implicating spatial receptor organization as a determinant of infection. This study investigates how HIV-1 Env (specifically X4-tropic gp120) alters the membrane organization and dynamics of the chemokine receptor CXCR4 and its WHIM-associated mutant, CXCR4R334X, in a CD4-dependent manner. Using single-particle tracking total internal reflection fluorescence microscopy (SPT-TIRF-M), the authors demonstrate that both soluble gp120 and virus-like particles (VLPs) displaying gp120 induce CXCR4 nanoclustering, reduce receptor diffusivity, and promote immobile nanoclusters of CXCR4 at the membrane of Jurkat T cells and primary CD4⁺ T cell blasts. The work offers new insights into the spatial organisation of receptors during HIV-1 entry and infection. The manuscript is well-written, and the findings are significant.

Significance:

Nature and significance of the advance:

This work marks a conceptual and mechanistic breakthrough in understanding HIV-1 entry. It goes beyond the static view of Env-co-receptor interaction to show that nanoscale reorganization of CXCR4, distinct from chemokine-induced clustering, occurs during HIV-1 Env engagement and may be essential for infection.

Context within existing literature. Previous studies established Env-induced CD4 clustering (Yin et al., 2020) and chemokine-induced CXCR4 nanocluster formation (Martínez-Muñoz et al., 2018), but the exact nanoscale rearrangement of CXCR4 in the context of HIV-1 Env and physiological Env densities remains unquantified. This study addresses this gap using SPT-TIRF, STED microscopy, and functional assays.

Audience and influence: The findings will be of interest to researchers in HIV virology, membrane receptor biology, viral entry mechanisms, and therapeutic target development. The receptor-clustering aspect could also influence broader fields of study, such as GPCR organization and immune receptor signalling.

Reviewer expertise: I can evaluate HIV-1 entry mechanisms, viral glycoprotein-host-host-host receptor interactions, single-molecule fluorescence microscopy, and membrane protein dynamics. I am less equipped to evaluate the deep structural modelling aspects, though the in silico AlphaFold results are straightforward to interpret in context.

---

## [Referee Report · Reviewer #3 (Public review)]

Summary:

The authors investigate how HIV-1 Env engagement affects the nanoscale organization and dynamics of the CXCR4 coreceptor on target cells. Using single-particle tracking TIRF microscopy, they analyze CXCR4 distribution following exposure to gp120 or HIV virus-like particles, including both wild-type CXCR4 and the WHIM-associated CXCR4.R334X variant. The study further examines the role of CD4-CXCR4 heterodimerization and contrasts Env-induced receptor organization with that elicited by the natural ligand CXCL12.

Evaluation:

A major strength of this work is the integration of high-resolution imaging with functional and comparative analyses that distinguish Env-induced CXCR4 clustering from chemokine-driven effects. The experiments are clearly described, include appropriate controls, and are supported by quantitative analyses that are consistent across experiments. The revised manuscript appears to have addressed many of the technical and interpretive issues raised during initial review, improving clarity around data analysis and strengthening confidence in the conclusions.

I am not an expert in TIRF microscopy or single-molecule tracking and defer to other reviewers regarding limits of imaging and tracking methods. However, I did not identify major inconsistencies between the biological data presented and the conclusions drawn.

The authors data support the conclusion that HIV-1 Env, delivered as gp120 or virus-like particles, promotes CD4-dependent nanoscale clustering of CXCR4, including the CXCR4.R334X variant associated with WHIM syndrome, in a manner distinct from CXCL12-induced receptor organization. The authors are generally careful to frame their conclusions in proportion to the evidence and avoid overinterpretation.

Overall, this study builds on prior work on CXCR4 distribution and HIV entry by providing higher-resolution insight into receptor nanoclustering and its modulation by Env. The findings provide a mechanistic refinement rather than a conceptual paradigm shift but is a valuable dataset useful to researchers studying HIV entry, coreceptor biology, and membrane receptor organization.

Reviewer expertise: HIV-1 Envelope glycoproteins and entry assays, HIV broadly neutralizing antibodies, HIV vaccine design

Comments on revised version:

This reviewer has no further recommendations and thanks the authors for clarifying that the Env content in gp120-VLPs was lower than the NL4-3deltaIN particles but that the percentage of mature particles in the gp120-VLPs was higher.

---

## [Referee Report · Reviewer #4 (Public review)]

Summary:

The authors investigate the impact of surface bound HIV gp120 and VLPs on CXCR4 dynamics in Jurkat T cells expressing WT or WHIM syndrome mutated CXCR4, which has a defective response to CXCL12. Jurkat cells were transfected with CXCR4-AcGFP. Images were acquired and a single particle tracking routine was applied to generate information about nanoclustering and diffusion, and FRET was used to investigate CD4-CXCR4 proximity. They compare effects of soluble gp120 to immature and mature VLPs, which include varying degrees of gp120 clustering. They find that solid phase gp120 or VLP can increase CXCR4 clustering size and decrease diffusion in Jurkat cells. Surprisingly, VLP lacking gp120 could increase CXCR4 clustering and speed, which is paradoxical as there were no known ligands on the VLPs, but they likely carry many cellular proteins with potential interactions. The impact of CXCL12 and gp120 binding to CXCR4 was different in terms of clustering and receptor down-regulation.

Significance:

The strengths are that it's an important question and the reagents are well prepared and characterised. They are detecting quantitative effects that will likely be reproducible. The information generated is potentially useful for those studying HIV infection processes and strategies to prevent infection.

The major weakness is that the conditions for the SPT experiments are not ideal in that the density of particles is too high for SPT and the single molecule basis for assessing nanoclusters is not clear. This means that the data is getting at complex molecules phenomena and less likely be generating pure single molecules measurements.

Comments on revised version:

The authors should make the tracking data available and this will aid others in following up on it.

---

## [Author Response]

**Point-by-point description of the revisions**

**Reviewer #1:**

Thank you very much for considering that our manuscript evaluates an important question and that the reagents used are well prepared and characterized. We also much appreciate that you consider the information generated as potentially useful for those studying HIV infection processes and strategies to prevent infection.

(1) While a single particle tracking routine was applied to the data, it's not clear how the signal from a single GFP was defined and if movement during the 100 ms acquisition time impacts this. My concern would be that the routine is tracking fluctuations, and these are related to single particle dynamics, it appears from the movies that the density or the GFP tagged receptors in the cells is too high to allow clear tracking of single molecules. SPT with GFP is very difficult due to bleaching and relatively low quantum yield. Current efforts in this direction that are more successful include using SNAP tags with very photostable organic fluorophores. The data likely does mean something is happening with the receptor, but they need to be more conservative about the interpretation.Some of the paradoxical effects might be better understood through deeper analysis of the SPT data, particularly investigation of active transport and more detailed analysis of "immobile" objects. Comments on early figures illustrate how this could be approached. This would require selecting acquisitions where the GFP density is low enough for SPT and performing a more detailed analysis, but this may be difficult to do with GFP.When the authors discuss clusters of <2 or >3, how do they calibrate the value of GFP and the impact of diffusion on the measurement. One way to approach this might be single molecules measurements of dilute samples on glass vs in a supported lipid bilayer to map the streams of true immobility to diffusion at >1 µm2/sec.

We fully understand the reviewer’s apprehensions regarding the application of these high-end biophysical techniques, in particular the associated complexity of the data analysis. We provide below extensive explanations on our methodology, which we hope will satisfactorily address all of the reviewer’s concerns.

We would first like to emphasize that the experimental conditions and the quantitative analysis used in our current experiments are similar to the established protocols and methodologies applied by our group previously (Martinez-Muñoz et al. Mol. Cell, 2018; García-Cuesta et al. PNAS, 2022; Gardeta et al. Frontiers in Immunol., 2022; García-Cuesta et al. eLife, 2024; Gardeta et al. Cell. Commun. Signal., 2025) and by others (Calebiro et al. PNAS, 2013; Jaqaman et al. Cell, 2011; Mattila et al. Immunity, 2013; Torreno-Pina et al. PNAS, 2014; Torreno-Pina et al. PNAS, 2016).

As SPT (single-particle tracking) experiments require low-expressing conditions in order to follow individual trajectories (Manzo & García-Parajo Rep. Prog. Phys., 2015), we transiently transfected Jurkat CD4^+^ cells with CXCR4-AcGFP or CXCR4^R334X^-AcGFP. At 24 h post-transfection, cells expressing low CXCR4-AcGFP levels were selected by a MoFlo Astrios Cell Sorter (BeckmanCoulter) to ensure optimal conditions for SPT. Using Dako Qifikit (DakoCytomation), we quantified the number of CXCR4 receptors and found ~8,500 – 22,000 CXCR4-AcGFP receptors/cell, which correspond to a particle density ~2 – 4.5 particles/µm^2^ (Author response image 1) and are similar to the expression levels found in primary human lymphocytes.

**Author response image 1. sa5fig1:** Purified AcGFP monomeric protein was immobilized on glass at various concentrations. Dependency of the distribution of particle components on particle density was calculated; >95% were monomeric single particles at 2.0-4.5 particles/µm^2^. This range of particle density was used to analyze the dynamics of CXCR4-AcGFP, or CXCR4^R334X^-AcGFP single particles on JKCD4 cells.

These cells were resuspended in RPMI supplemented with 2% FBS, NaPyr and L-glutamine and plated on 96-well plates for at least 2 h. Cells were centrifuged and resuspended in a buffer with HBSS, 25 mM HEPES, 2% FBS (pH 7.3) and plated on glass-bottomed microwell dishes (MatTek Corp.) coated with fibronectin (FN) (Sigma-Aldrich, 20 µg/ml, 1 h, 37°C). To observe the effect of the ligand, we coated dishes with FN + CXCL12; FN + X4-gp120 or FN + VLPs, as described in material and methods; cells were incubated (20 min, 37°C, 5% CO_2_) before image acquisition.

For SPT measurements, we use a total internal reflection fluorescence (TIRF) microscope (Leica AM TIRF inverted) equipped with an EM-CCD camera (Andor DU 885-CS0-#10-VP), a 100x oilimmersion objective (HCX PL APO 100x/1.46 NA) and a 488-nm diode laser. The microscope was equipped with incubator and temperature control units; experiments were performed at 37°C with 5% CO_2_. To minimize photobleaching effects before image acquisition, cells were located and focused using the bright field, and a fine focus adjustment in TIRF mode was made at 5% laser power, an intensity insufficient for single-particle detection that ensures negligible photobleaching. Image sequences of individual particles (500 frames) were acquired at 49% laser power with a frame rate of 10 Hz (100 ms/frame). The penetration depth of the evanescent field used was 90 nm.

We performed automatic tracking of individual particles using a very well established and common algorithm first described by Jaqaman (Jaqaman et al. Nat. Methods, 2008). Nevertheless, we would stress that we implemented this algorithm in a supervised fashion, i.e., we visually inspect each individual trajectory reconstruction in a separate window. Indeed, this algorithm is not able to quantify merging or splitting events.

We follow each individual fluorescence spot frame-by-frame using a three-by-three matrix around the centroid position of the spot, as it diffuses on the cell membrane. To minimize the effect of photon fluctuations, we averaged the intensity over 20 frames. Nevertheless, to assure the reviewer that most of the single molecule traces last for at least 50 frames (i.e., 5 seconds), we provide the following data and arguments. We currently measure the photobleaching times from individual CD86-AcGFP spots exclusively having one single photobleaching step to guarantee that we are looking at individual CD86-AcGFP molecules. The distribution of the photobleaching times is shown below (Author response image 2). Fitting of the distribution to a single exponential decay renders a t0 value of ~5 s. Thus, with 20 frames averaging, we are essentially measuring the whole population of monomers in our experiments. As the survival time of a molecule before photobleaching will strongly depend on the excitation conditions, we used low excitation conditions (2 mW laser power, which corresponds to an excitation power density of ~0.015 kW/cm^2^ considering the illumination region) and longer integration times (100 ms/frame) to increase the signal-to-background for single GFP detection while minimizing photobleaching.

**Author response image 2. sa5fig2:** Single molecule photobleaching times measured directly from single molecule trajectories of CD86-AcGFP, considering only traces that exhibit single molecule photobleaching steps. The experimental data are shown in gray bars (n=273 trajectories over 3 independent experiments). The red line corresponds to a single exponential decay fitting of the experimental data, from where t_o_ has been extracted.

To infer the stoichiometry of receptor complexes, we also perform single-step photobleaching analysis of the TIRF trajectories to establish the existence of different populations of monomers, dimers, trimers and nanoclusters and extract their percentage. Some representative trajectories of CXCR4-AcGFP with the number of steps detected are shown in new Supplementary Figure 1.

The emitted fluorescence (arbitrary units, a.u.) of each spot in the cells is quantified and normalized to the intensity emitted by monomeric CD86-AcGFP spots that strictly showed a single photobleaching step (Dorsch et al. Nat. Methods, 2009). We have preferred to use CD86-AcGFP in cells rather than AcGFP on glass to exclude any potential effect on the different photodynamics exhibited by AcGFP when bound directly to glass. We have also previously shown pharmacological controls to exclude CXCL12-mediated receptor clustering due to internalization processes (Martinez-Muñoz et al. Mol. Cell, 2018) that, together with the evaluation of single photobleaching steps and intensity histograms, allow us to exclude the presence of vesicles in our data. Thus, the dimers, trimers and nanoclusters found in our data do correspond to CXCR4 molecules on the cell surface. Finally, distribution of monomeric particle intensities, obtained from the photobleaching analysis, was analyzed by Gaussian fitting, rendering a mean value of 980 ± 86 a.u. This value was then used as the monomer reference to estimate the number of receptors per particle in both cases, CXCR4-AcGFP and CXCR4^R334X^-AcGFP (new Supplementary Figure 1).

(2) I understand that the CXCL12 or gp120 are attached to the substrate with fibronectin for adhesion. I'm less clear how how that VLPs are integrated. Were these added to cells already attached to FN?

For TIRF-M experiments, cells were adhered to glass-bottomed microwell dishes coated with fibronectin, fibronectin + CXCL12, fibronectin + X4-gp120, or fibronectin + VLPs. As for CXCL12 and X4-gp120, the VLPs were attached to fibronectin taking advantage of electrostatic interactions. To clarify the integration of the VLPs in these assays, we have stained the microwell dishes coated with fibronectin and those coated with fibronectin + VLPs with wheat germ agglutinin (WGA) coupled to Alexa647 (Author response image 3) and evaluated the staining by confocal microscopy. These results indicate the presence of carbohydrates on the VLPs and are, therefore, indicative of the presence of VLPs on the fibronectin layer.

**Author response image 3. sa5fig3:** Representative confocal images of microwell dishes coated with fibronectin ((left panel) or fibronectin + VLPs (right panel)) and stained with wheat germ agglutinin (WGA) coupled to Alexa647. Bar scale 1µm.

Moreover, it is important to remark that the effect of the VLPs on CXCR4 behavior at the cell surface observed by TIRF-M confirmed that the VLPs remained attached to the substrate during the experiment.

(3) Fig 1A - The classification of particle tracks into mobile and immobile is overly simplistic description that goes back to bulk FRAP measurements and it not really applicable to single molecule tracking data, where it's rare to see anything that is immobile and alive. An alternative classification strategy uses sub-diffusion, normal diffusion and active diffusion (or active transport) to descriptions and particles can transition between these classes over the tracking period. Fig 1B- this data might be better displayed as histograms showing distributions within the different movement classes.

In agreement with the reviewer’s commentary, the majority of the particles detected in our TIRFM experiments were indeed mobile. However, we also detected a variable, and biologically appreciable, percentage of immobile particles depending on the experimental condition analyzed (Figure 1A in the main manuscript). To establish a stringent threshold for identifying these immobile particles under our specific experimental conditions, we used purified monomeric AcGFP proteins immobilized on glass coverslips. Our analysis demonstrated that 95% of these immobilized proteins showed a diffusion coefficient £0.0015 µm^2^/s; consequently, this value was established as the cutoff to distinguish immobile from mobile trajectories. While the observation of truly immobile entities in a dynamic, living system is rare, the presence of these particles under our conditions is biologically significant. For instance, the detection of large, immobile receptor nanoclusters at the plasma membrane is entirely consistent with facilitating key cellular processes, such as enabling the robust signaling cascade triggered by ligand binding or promoting the crucial events required for efficient viral entry into the cells.

Regarding the mobile receptors (defined as those with D_1-4_ values exceeding 0.0015 µm^2^/s), we observed distinct diffusion profiles derived from mean square displacement (MSD) plots (Figure V) (Manzo & García-Parajo Rep. Prog. Phys., 2015), which were further classified based on motion, using the moment scaling spectrum (MSS) (Ewers et al. PNAS, 2005). Under all experimental conditions, the majority of mobile particles, ~85%, showed confined diffusion: for example under basal conditions, without ligand addition, ~90% of mobile particles showed confined diffusion, ~8.5% showed Brownian-free diffusion and ~1.5% exhibited directed motion (new Supplementary Figure 5A in the main manuscript). These data have been also included in the revised manuscript to show, in detail, the dynamic parameters of CXCR4.

Due to the space constraints, it is very difficult to include all the figures generated. However, to ensure comprehensive assessment and transparency (for the purpose of this review), we have included below representative plots of the MSD values as a function of time from individual trajectories, showing different types of motion obtained in our experiments (Author response image 4).

**Author response image 4. sa5fig4:** Representative MSD plots from individual trajectories of CXCR4AcGFP detected by SPT-TIRF in resting JKCD4 cells showing different types of motion: (A) confined, (B) Brownian/Free, (C) direct transport.

(4) Fig 1C,D - It would be helpful to see a plot of D vs MSI at a single particle level. In comparing C and D I'm surprised there is not a larger difference between CXCL12 and X4-gp120. It would also be very important to see the behaviour of X4-gp120 on the CXCR4 deficient Jurkat that would provide a picture of CD4 diffusion. The CXCR4 nanoclustering related to the X4-gp120 could be dominated by CD4 behaviour.

As previously described, all analyses were performed under SPT conditions (see previous response to point 1). Figure 1C details the percentage of oligomers (>3 receptors/particle) calibrated using Jurkat CD4^+^ cells electroporated with monomeric CD86-AcGFP (Dorsch et al. Nat. Methods, 2009). The monomer value was determined by analyzing photobleaching steps as described in our previous response to point 1.

In our experiments, we observed a trend towards a higher number of oligomers upon activation with CXCL12 compared with X4-gp120. This trend was further supported by measurements of Mean Spot Intensity. However, the values are also influenced by the number of larger spots, which represents a minor fraction of the total spots detected.

The differences between the effect triggered by CXCL12 or X4-gp120 might also be attributed to a combination of factors related to differences in ligand concentration, their structure, and even to the technical requirements of TIRF-M. Both ligands are in contact with the substrate (fibronectin) and the specific nature of this interaction may differ between both ligands and influence their accessibility to CXCR4. Moreover, the requirement of the prior binding of gp120 to CD4 before CXCR4 engagement, in contrast to the direct binding of CXCL12 to CXCR4, might also contribute to the differences observed.

We previously reported that CXCL12-mediated CXCR4 dynamics are modulated by CD4 coexpression (Martinez-Muñoz et al. Mol. Cell, 2018). We have now detected the formation of CD4 heterodimers with both CXCR4 and CXCR4^R334X^, and found that these conformations are influenced by gp120-VLPs. In the present manuscript, we did not focus on CD4 clustering as it has been extensively characterized previously (Barrero-Villar et al. J. Cell Sci., 2009; JiménezBaranda et al. Nat. Cell. Biol., 2007; Yuan et al. Viruses, 2021). Regarding the investigation of the effects of X4-gp120 on CXCR4-deficient Jurkat cells, which would provide a picture of CD4 diffusion, we would note that a previous report has already addressed this issue using single molecule super-resolution imaging, and revealed that CD4 molecules on the cell membrane are predominantly found as individual molecules or small clusters of up to 4 molecules, and that the size and number of these clusters increases upon virus binding or gp120 activation (Yuan et al. Viruses, 2021).

(5) Fig S1D- This data is really interesting. However, if both the CD4 and the gp120 have his tags they need to be careful as poly-His tags can bind weakly to cells and increasing valency could generate some background. So, they should make the control is fair here. Ideally, using non-his tagged person of sCD4 and gp120 would be needed ideal or they need a His-tagged Fab binding to gp120 that doesn't induce CXCR4 binding.

New Supplementary Figure 2D shows that X4-gp120 does not bind Daudi cells (these cells do not express CD4) in the absence of soluble CD4. While the reviewer is correct to state that both proteins contain a Histidine Tag, cell binding is only detected if X4-gp120 binds sCD4. Nonetheless, we have included in the revised Supplementary Figure 2D a control showing the negative binding of sCD4 to Daudi cells in the absence of X4-gp120. Altogether, these results confirm that only sCD4/X4-gp120 complexes bind these cells.

(6) Fig S4- Panel D needs a scale bar. I can't figure out what I'm being shown without this.

Apologies. A scale bar has been included in this panel (new Supplementary Figure 6D).

**Reviewer #2:**
(1) This study is well described in both the main text and figures. Introduction provides adequate background and cites the literature appropriately. Materials and Methods are detailed. Authors are careful in their interpretations, statistical comparisons, and include necessary controls in each experiment. The Discussion presents a reasonable interpretation of the results. Overall, there are no major weaknesses with this manuscript.

We very much appreciate the positive comments of the reviewer regarding the broad interest and strength of our work.

(2) NL4-3deltaIN and immature HIV virions are found to have less associated gp120 relative to wild-type particles. It is not obvious why this is the case for the deltaIN particles or genetically immature particles. Can the authors provide possible explanations? (A prior paper was cited, Chojnacki et al Science, 2012 but can the current authors provide their own interpretation.)

Our conclusion from the data is actually exactly the opposite. As shown in Figure 2D, the gp120 staining intensity was higher for NL4-3DIN particles (1,786 a.u.) than for gp120-VLPs (1,223 a.u.), indicating lower expression of Env proteins in the latter. Furthermore, analysis of gp120 intensity per particle (Figure 2E) confirmed that gp120-VLPs contained fewer gp120 molecules per particle than NL4-3DIN virions. These levels were comparable with, or even lower than, those observed in primary HIV-1 viruses (Zhu et al. Nature, 2006). This reduction was a direct consequence of the method used to generate the VLPs, as our goal was to produce viral particles with minimal gp120 content to prevent artifacts in receptor clustering that might occur using high levels of Env proteins in the VLPs to activate the receptors.

This misunderstanding may arise from the fact that we also compared Gag condensation and Env distribution on the surface of gp120-VLPs with those observed in genetically immature particles and integrase-defective NL4-3ΔIN virions, which served as controls. STED microscopy data revealed differences in Env distribution between gp120-VLPs and NL4-3ΔIN virions, supporting the classification of gp120-VLPs as mature particles (Figure 2 A,B).

**Reviewer #3**:

We thank the reviewer for considering that our work offers new insights into the spatial organization of receptors during HIV-1 entry and infection and that the manuscript is well written, and the findings significant.

(1) For mechanistic basis of gp120-CXCR4 versus CXCL12-CXCR4 differences. Provide additional structural or biochemical evidence to support the claim that gp120 stabilises a distinct CXCR4 conformation compared to CXCL12. If feasible, include molecular modelling, mutagenesis, or crosslinking experiments to corroborate the proposed conformational differences.

We appreciate the opportunity to clarify this point. The specific claim that gp120 stabilizes a conformation of CXCR4 that is distinct from the CXCL12-bound state was not explicitly stated in our manuscript, although we agree that our data strongly support this possibility. It is important to consider that CXCL12 binds directly to CXCR4, whereas gp120 requires prior sequential binding to CD4, and its subsequent interaction is with a CXCR4 molecule that is already forming part of the CD4/CXCR4 complex, as demonstrated by our FRET experiments and supported by previous studies (Zaitseva et al. J. Leuk. Biol., 2005; Busillo & Benovic Biochim. Biophys. Acta, 2007; Martínez-Muñoz et al. PNAS, 2014). This difference makes it inherently complex to compare the conformational changes induced by gp120 and CXCL12 on CXCR4.

However, our findings show that both stimuli induce oligomerization of CXCR4, a phenomenon not observed when mutant CXCR4^R334X^ was exposed to the chemokine CXCL12 (García-Cuesta et al. PNAS, 2022).

(1) CXCL12 induced oligomerization of CXCR4 but did not affect the dynamics of CXCR4^R334X^ (Martinez-Muñoz et al. Mol. Cell, 2018; García-Cuesta et al. PNAS, 2022). By contrast, X4-gp120 and the corresponding VLPs—which require initial binding to CD4 to engage the chemokine receptor—stabilized oligomers of both CXCR4 and CXCR4^R334X^.

(2) FRET analysis revealed distinct FRET_50_ values for CD4/CXCR4 (2.713) and CD4/CXCR4^R334X^ (0.399) complexes, suggesting different conformations for each complex.

(3) Consistent with previous reports (Balabanian et al. Blood, 2005; Zmajkovicova et al. Front. Immunol., 2024; García-Cuesta et al. PNAS, 2022), the molecular mechanisms activated by CXCL12 are distinct when comparing CXCR4 with CXCR4^R334X^. For instance, CXCL12 induces internalization of CXCR4, but not of mutant CXCR4^R334X^. Conversely, X4-gp120 triggers approximately 25% internalization of both receptors. Similarly, CXCL12 does not promote CD4 internalization in cells co-expressing CXCR4 or CXCR4^R334X^, whereas X4-gp120 does, although CD4 internalization was significantly higher in cells co-expressing CXCR4.

These findings suggest that CD4 influences the conformation and the oligomerization state of both co-receptors. To further support this hypothesis, we have conducted new in silico molecular modeling of CD4 in complex with either CXCR4 or its mutant CXCR4^R334X^ using AlphaFold 3.0 (Abramson et al. Nature, 2024). The server was provided with both sequences, and the interaction between the two molecules for each protein was requested. It produced a number of solutions, which were then analyzed using the software ChimeraX 1.10 (Meng et al. Protein Sci., 2023). CXCR4 and its mutant, CXCR4^R334X^ bound to CD4, were superposed using one of the CD4 molecules from each complex, with the aim of comparing the spatial positioning of CD4 molecules when interacting with CXCR4.

**Author response image 5. sa5fig5:** CD4/CXCR4 complexes were superimposed with CD4/CXCR4 complexes (left panel) or CD4/CXCR4^R334X^ complexes (right panels). Arrows indicate the CD4 molecule used as reference for the superimposing.

As illustrated in Author response image 5, the superposition of the CD4/CXCR4 complexes was complete. However, when CD4/CXCR4 complexes were superimposed with CD4/CXCR4^R334X^ complexes using the same CD4 molecule as a reference, indicated by an arrow in the figure, a clear structural deviation became evident. The main structural difference detected was the positioning of the CD4 transmembrane domains when interacting with either the wild-type or mutant CXCR4. While in complexes with CXCR4, the angle formed by the lines connecting residues E416 at the C-terminus end of CD4 with N196 in CXCR4 was 12°, for the CXCR4^R334X^ complex, this angle increased to 24°, resulting in a distinct orientation of the CD4 extracellular domain (Author response image 6).

**Author response image 6. sa5fig6:** Comparison of the angle between the transmembrane domains of CD4 in CXCR4 WT and WHIM complexes. The angle between residues N196 from one CXCR4 molecule and E416 from the two CD4 dimer molecules was calculated for the CXCR4 WT (12°) and WHIM (24°) complexes to demonstrate the difference in CD4 positioning.

To further analyze the models obtained, we employed PDBsum software (Laskowski & Thornton Protein Sci., 2021) to predict the CD4/CXCR4 interface residues. Data indicated that at least 50% of the interaction residues differed when the CD4/CXCR4 interaction surface was compared with that of the CD4/CXCR4^R334X^ complex (Author response image 7). It is important to note that while some hydrogen bonds were present in both complex models, others were exclusive to one of them. For instance, whereas Cys^394^(CD4)-Tyr^139^ and Lys^299^(CD4)-Glu^272^ were present in both CD4/CXCR4 and CD4/CXCR4^R334X^ complexes, the pairs Asn^337^(CD4)-Ser^27^(CXCR4^R334X^) and Lys^325^(CD4)-Asp^26^(CXCR4^R334X^) were only found in CD4/CXCR4^R334X^ complexes.

**Author response image 7. sa5fig7:** Interacting residues at the CD4/CXCR4 interface. The panel displays the interface residues from the CXCR4 and CD4 oligomer. CD4 residues labeled with a red sphere show the interacting residues present in both CXCR4-WT and –WHIM hetero- oligomers. The continuous red lines represent a saline bridge, while the blue lines indicate a hydrogen bond and the dashed red lines represent non-bonded interactions. As illustrated in the figure, half of the interacting residues differ between the WT and WHIM models, indicating that the interacting surfaces are also distinct.

These findings, which are consistent with our FRET results, suggest distinct interaction surfaces between CD4 and the two chemokine receptors. Overall, these results are compatible with differences in the spatial conformation adopted by these complexes.

(2) For Empty VLP effects on CXCR4 dynamics: Explore potential causes for the observed effects of Envdeficient VLPs. It's valuable to include additional controls such as particles from non-producer cells, lipid composition analysis, or blocking experiments to assess nonspecific interactions.

As VLPs are complex entities, we thought that the relevant results should be obtained comparing the effects of Env(-) VLPs with gp120-VLPs. Therefore, we would first remark that regardless of the effect of Env(-) VLPs on CXCR4 dynamics, the most evident finding in this study is the strong effect of gp120-VLPs compared with control Env(-) VLPs. Nevertheless, regarding the effect of the Env(-) VLPs compared with medium, we propose several hypotheses. As several virions can be tethered to the cell surface via glycosaminoglycans (GAGs), we hypothesized that VLPs-GAGs interactions might indirectly influence the dynamics of CXCR4 and CXCR4^R334X^ at the plasma membrane. Additionally, membrane fluidity is essential for receptor dynamics, therefore VLPs interactions with proteins, lipids or any other component of the cell membrane could also alter receptor behavior. It is well known that lipid rafts participate in the interaction of different viruses with target cells (Nayak & Hu Subcell. Biochem., 2004; Manes et al. Nat. Rev. Immunol., 2003; Rioethmullwer et al. Biochim. Biophys. Acta, 2006) and both the lipid composition and the presence of co-expressed proteins modulate ligand-mediated receptor oligomerization (Gardeta et al. Frontiers in Immunol., 2022; Gardeta et al. Cell. Commun. Signal., 2025). We have thus performed Raster Image Correlation Spectroscopy (RICS) analysis to assess membrane fluidity through membrane diffusion measurements on cells treated with Env(-) VLPs.

Jurkat cells were labeled with Di-4-ANEPPDHG and seeded on FN and on FN + VLPs prior to analysis by RICS on confocal microscopy. The results indicated no significant differences in membrane diffusion under the treatment tested, thereby discarding an effect of VLPs on overall membrane fluidity (Author response image 8).

**Author response image 8. sa5fig8:** VLPs treatment does not alter cell membrane fluidity. Diffusion values obtained by RICS from JKCD4X4 cells. (n = 3, with at least 10 cells analyzed per experiment and condition; n.s., not significant).

Nonetheless, these results do not rule out other non-specific interactions of Env(-) VLPs with membrane proteins that could affect receptor dynamics. For instance, it has been reported that Ctype lectin DC-SIGN acts as an efficient docking site for HIV-1 (Cambi et al. J. Cell. Biol., 2004; Wu & KewalRamani Nat. Rev. Immunol., 2006). However, a detailed investigation of these possible mechanisms is beyond the scope of this manuscript.

(3) For Direct link between clustering and infection efficiency - Test whether disruption of CXCR4 clustering (e.g., using actin cytoskeleton inhibitors, membrane lipid perturbants, or clustering-deficient mutants) alters HIV-1 fusion or infection efficiency.

Designing experiments using tools that disrupt receptor clustering by interacting with the receptors themselves is difficult and challenging, as these tools bind the receptor and can therefore alter parameters such as its conformation and/or its distribution at the cell membrane, as well as affect some cellular processes such as HIV-1 attachment and cell entry. Moreover, effects on actin polymerization or lipids dynamics can affect not only receptor clustering but also impact on other molecular mechanisms essential for efficient infection.

Many previous reports have, nonetheless, indirectly correlated receptor clustering with cell infection efficiency. Cholesterol plays a key role in the entry of several viruses. Its depletion in primary cells and cell lines has been shown to confer strong resistance to HIV-1-mediated syncytium formation and infection by both CXCR4- and CCR5-tropic viruses (Liao et al. AIDS Res. Hum. Retroviruses, 2021). Moderate cholesterol depletion also reduces CXCL12-induced CXCR4 oligomerization and alters receptor dynamics (Gardeta et al. Cell. Commun. Signal., 2025). By restricting the lateral diffusion of CD4, sphingomyelinase treatment inhibits HIV-1 fusion (Finnegan et al. J. Virol., 2007). Depletion of sphingomyelins also disrupts CXCL12mediated CXCR4 oligomerization and its lateral diffusion (Gardeta et al. Front Immunol., 2022). Additional reports highlight the role of actin polymerization at the viral entry site, which facilitates clustering of HIV-1 receptors, a crucial step for membrane fusion (Serrano et al. Biol. Cell., 2023). Blockade of actin dynamics by Latrunculin A treatment, a drug that sequesters actin monomers and prevents its polymerization, blocks CXCL12-induced CXCR4 dynamics and oligomerization (Martínez-Muñoz et al. Mol. Cell, 2018).

Altogether, these findings strongly support our hypothesis of a direct link between CXCR4 clustering and the efficiency of HIV-1 infection.

(4) CD4/CXCR4 co-endocytosis hypothesis - Support the proposed model with direct evidence from livecell imaging or co-localization experiments during viral entry. Clarification is needed on whether internalization is simultaneous or sequential for CD4 and CXCR4.

When referring to endocytosis of CD4 and CXCR4, we only hypothesized that HIV-1 might promote the internalization of both receptors either sequentially or simultaneously. The hypothesis was based in several findings:

a) Previous studies have suggested that HIV-1 glycoproteins can reduce CD4 and CXCR4 levels during HIV-1 entry (Choi et al. Virol. J., 2008; Geleziunas et al. FASEB J, 1994; Hubert et al. Eur. J. Immunol., 1995).

b) Receptor endocytosis has been proposed as a mechanism for HIV-1 entry (Daecke et al. J. Virol., 2005; Aggarwal et al. Traffick, 2017; Miyauchi et al. Cell, 2009; Carter et al. Virology, 2011).

c) Our data from cells activated with X4-gp120 demonstrated internalization of CD4 and chemokine receptors, which correlated with HIV-1 infection in PBMCs from WHIM patients and healthy donors.

d) CD4 and CXCR4 have been shown to co-localize in lipid rafts during HIV-1 infection (Manes et al. EMBO Rep., 2000; Popik et al. J. Virol., 2002)

e) Our FRET data demonstrated that CD4 and CXCR4 form heterocomplexes and that FRET efficiency increased after gp120-VLPs treatment.

We agree with the reviewer that further experiments are required to test this hypothesis, however, we believe that this is beyond the scope of the current manuscript.

Minor Comments:(1) The conclusions rely solely on the HXB2 X4-tropic Env. It would strengthen the study to assess whether other X4 or dual-tropic strains induce similar receptor clustering and dynamics.

The primary goal of our current study was to investigate the dynamics of the co-receptor CXCR4 during HIV-1 infection, motivated by previous reports showing CD4 oligomerization upon HIV1 binding and gp120 stimulation (Yuan et al. Viruses, 2021). We initially used a recombinant X4gp120, a soluble protein that does not fully replicate the functional properties of the native HIV-1 Env. Previous studies have shown that Env consists of gp120 trimers, which redistribute and cluster on the surface of virions following proteolytic Gag cleavage during maturation (Chojnacki et al. Nat. Commun., 2017). An important consideration in receptor oligomerization studies is the concentration of recombinant gp120 used, as it does not accurately reflect the low number of Env trimers present on native HIV-1 particles (Hart et al. J. Histochem. Cytochem., 1993; Zhu et al. Nature, 2006). To address these limitations, we generated virus-like particles (VLPs) containing low levels of X4-gp120 and repeated the dynamic analysis of CXCR4. The use of primary HIV-1 isolates was limited, in this project, to confirm that PBMCs from both healthy donors and WHIM patients were equally susceptible to infection. This result using a primary HIV-1 virus supports the conclusion drawn from our in vitro approaches. We thus believe that although the use of other X4- and dual-tropic strains may complement and reinforce the analysis, it is far beyond the scope of the current manuscript.

(2) Given the observed clustering effects, it would be valuable to explore whether gp120-induced rearrangements alter epitope exposure to broadly neutralizing antibodies like 17b or 3BNC117. This would help connect the mechanistic insights to therapeutic relevance.

As 3BNC117, VRC01 and b12 are broadly neutralizing mAbs that recognize conformational epitopes on gp120 (Li et al. J. Virol., 2011; Mata-Fink et al. J. Mol. Biol., 2013), they will struggle to bind the gp120/CD4/CXCR4 complex and therefore may not be ideal for detecting changes within the CD4/CXCR4 complex. The experiment suggested by the reviewer is thus challenging but also very complex. It would require evaluating antibody binding in two experimental conditions, in the absence and in the presence of oligomers. However, our data indicate that receptor oligomerization is promoted by X4-gp120 binding, and the selected antibodies are neutralizing mAbs, so they should block or hinder the binding of gp120 and, consequently, receptor oligomerization. An alternative approach would be to study the neutralizing capacity of these mAbs on cells expressing CD4/CXCR4 or CD4/CXCR4^R334X^ complexes. Variations in their neutralizing activity could be then extrapolated to distinct gp120 conformations, which in turn may reflect differences between CD4/CXCR4 and CD4/CXCR4^R334X^ complexes.

We thus assessed the ability of the VRC01 and b12, anti-gp120 mAbs, which were available in our laboratory, to neutralize gp120 binding on cells expressing CD4/CXCR4 or CD4/CXCR4^R334X^. Specifically, increasing concentrations of each antibody were preincubated (60 min, 37ºC) with a fixed amount of X4-gp120 (0.05 µg/ml). The resulting complexes were then incubated with Jurkat cells expressing CD4/CXCR4 or CD4/CXCR4^R334X^ (30 min, 37ºC) and, finally, their binding was analyzed by flow cytometry. Although we did not observe statistically significant differences in the neutralization capacity of b12 or VRC01 for the binding of X4-gp120 depending on the presence of CXCR4 or CXCR4^334X^, we observed a trend for greater concentrations of both mAbs to neutralize X4-gp120 binding in Jurkat CD4/CXCR4 cells than in Jurkat CD4/CXCR4^R334X^ cells (Author response image 9).

**Author response image 9. sa5fig9:** Flow cytometry analysis of gp120 binding to Jurkat cells expressing CD4/CXCR4 or CD4/CXCR4^R334X^ in the presence of different concentrations of the neutralizing anti-gp120 antibodies b12 (left panel) and VRC01 (right panel). AUC comparison by Welch’s t-test: pvalues 0.2950 and 0.2112 for b12 and VRC01 respectively (n = 2).

These slight alterations in the neutralizing capacity of b12 and VRC01 mAbs may thus suggest minimal differences in the conformations of gp120 depending of the coreceptor used. We also detected that X4-gp120 and VLPs expressing gp120, which require initial binding to CD4 to engage the chemokine receptor, stabilized oligomers of both CXCR4 and CXCR4^R334X^, but FRET data indicated distinct FRET_50_ values between the partners, (2.713) for CD4/CXCR4 and (0.399) for CD4/CXCR4^R334X^ (Figure 5A,B in the main manuscript). Moreover, we also detected significantly more CD4 internalization mediated by X4-gp120 in cells co-expressing CD4 and CXCR4 than in those co-expressing CD4 and CXCR4^R334X^ (Figure 6 in the main manuscript). Overall these latter data and those included in Author response images 5,6 and 7 indicate distinct conformations within each receptor complexes.

(3) TIRF imaging limits analysis to the cell substrate interface. It would be useful to clarify whether CXCR4 receptor clustering occurs elsewhere, such as at immunological synapses or during cell-to-cell contact.

In recent years, chemokine receptor oligomerization has gained significant research interest due to its role in modulating the ability of cells to sense chemoattractant gradients. This molecular organization is now recognized as a critical factor in governing directed cell migration (Martínez-Muñoz et al. Mol. Cell, 2018; García-Cuesta et al. PNAS, 2022, Hauser et al. Immunity, 2016). In addition, advanced imaging techniques such as single-molecule and super-resolution microscopy have been used to investigate the spatial distribution and dynamic behaviour of CXCR4 within the immunological synapse in T cells (Felce et al. Front. Cell Dev. Biol., 2020). Building on these findings, we are currently conducting a project focused on characterizing CXCR4 clustering specifically within this specialized cellular region.

(4) In LVP experiments, it would be useful to report transduction efficiency (% GFP+ cells) alongside MSI data to relate VLP infectivity with receptor clustering functionally.

These experiments were designed to validate the functional integrity of the gp120 conformation on the LVPs, confirming their suitability for subsequent TIRF microscopy. Our objective was to establish a robust experimental tool rather than to perform a high-throughput quantification of transduction efficiency. It is for that reason that these experiments were included in new Supplementary Figure S6, which also contains the complete characterization of gp120-VLPs and LVPs. In such experimental conditions, quantifying the percentage of GFP-positive cells relative to the total number of cells plated in each well is very difficult. However, in line with the reviewer’s commentary and as we used the same number of cells in each experimental condition, we have included, in the revised manuscript, a complementary graph illustrating the GFP intensity (arbitrary units) detected in all the wells analyzed (new Supplementary Fig. 6E).

(5) To ensure that differences in fusion events (Figure 7B) are attributable to target cell receptor properties, consider confirming that effector cells express similar levels of HIV-1 Env. Quantifying gp120 expression by flow cytometry or western blot would rule out the confounding effects of variable Env surface density.

In these assays (Figure 7B), we used the same effector cells (cells expressing X4-gp120) in both experimental conditions, ensuring that any observed differences should be attributable solely to the target cells, either JKCD4X4 or JKCD4X4^R334X^. For this reason, in Figure 7A we included only the binding of X4-gp120 to the target cells which demonstrated similar levels of the receptors expressed by the cells.

(6) HIV-mediated receptor downregulation may occur more slowly than ligand-induced internalization. Including a 24-hour time point would help assess whether gp120 induces delayed CD4 or CXCR4 loss beyond the early effects shown and to better capture potential delayed downregulation induced by gp120.

The reviewer suggests using a 24-hour time point to facilitate detection of receptor internalization. However, such an extended incubation time may introduce some confounding factors, including receptor degradation, recycling and even de novo synthesis, which could affect the interpretation of the results. Under our experimental conditions, we observed that CXCL12 did not trigger CD4 internalization whereas X4-gp120 did. Interestingly, CD4 internalization depended on the coreceptor expressed by the cells.

(7) Increase label font size in microscopy panels for improved readability.

Of course; the font size of these panels has been increased in the revised version.

(8) Consider adding more references on ligand-induced co-endocytosis of CD4 and chemokine receptors during HIV-1 entry.

We have added more references to support this hypothesis (Toyoda et al. J. Virol., 2015; Venzke et al. J. Virol., 2006; Gobeil et al J. Virol., 2013).

(9) For Statistical analysis. Biological replicates are adequate, and statistical tests are generally appropriate. For transparency, report n values, exact p-values, and the statistical test used in every figure legend and discussed in the results.

Thank you for highlighting the importance of transparency in statistical reporting. We confirm that the n values for all experiments have been included in the figure legends. The statistical tests used for each analysis are also clearly indicated in the figure legends, and the interpretation of these results is discussed in detail in the Results section. Furthermore, the Methods section specifies the tests applied and the thresholds for significance, ensuring full transparency regarding our analytical approach.

In accordance with established conventions in the field, we have utilized categorical significance indicators (e.g., n.s., *, **, ***) within our figures to enhance readability and focus on biological trends. This approach is widely adopted in high-impact literature to prevent visual clutter. However, to ensure full transparency and reproducibility, we have ensured that the underlying statistical tests and thresholds are clearly defined in the respective figure legends and Methods section.

**Reviewer #4:**

We thank the reviewer for considering that this work is presented in a clear fashion, and the main findings are properly highlighted, and for remarking that the paper is of interest to the retrovirology community and possibly to the broader virology community.

We also agree on the interest that X4-gp120 clusters CXCR4^R334X^ suggests a different binding mechanism for X4-gp120 from that of the natural ligand CXCL12, an aspect that we are now evaluating. These data also indicate that WHIM patients can be infected by HIV-1 similarly to healthy people.

(1) The observation that "empty VLPs" reduce CXCR4 diffusivity is potentially interesting. However, it is not supported by the data owing to insufficient controls. The authors correctly discuss the limitations of that observation in the Discussion section (lines 702-704). However, they overinterpret the observation in the Results section (lines 509-512), suggesting non-specific interactions between empty VLPs, CD4 and CXCR4. I suggest either removing the sentence from the Results section or replacing it with a sentence similar to the one in the Discussion section.

In accordance with the reviewer`s suggestion, the sentence in the result section has been replaced with one similar to that found in the discussion section. In addition, we have performed Raster Image Correlation Spectroscopy (RICS) analysis using the Di-4-ANEPPDHQ lipid probe to assess membrane fluidity by means of membrane diffusion, and compared the results with those of cells treated with Env(-) VLPs. The results indicated that VLPs did not modulate membrane fluidity (Author response image 8). Nonetheless, these results do not rule out other potential non-specific interactions of the Env(-) VLPs with other components of the cell membrane that might affect receptor dynamics (see our response to point 2 of reviewer #3).

(2) In the case of the WHIM mutant CXCR4-R334X, the addition of "empty VLPs" did not cause a significant change in the diffusivity of CXCR4-R334X (Figure 4B). This result is in contrast with the addition of empty VLPs to WT CXCR4. However, the authors neither mention nor comment on that result in the results section. Please mention the result in the paper and comment on it in relation to the addition of empty VLPs to WT CXCR4.

We would remark that the main observation in these experiments should focus on the effect of gp120-VLPs, and the results indicates that gp120-VLPs promoted clustering of CXCR4 and of CXCR4^R334X^ and reduced their diffusion at the cell membrane. The Env(-) VLPs were included as a negative control in the experiments, to compare the data with those obtained using gp120VLPs. However, once we observed some residual effect of the Env(-) VLPs, we decided to give a potential explanation, formulated as a hypothesis, that the Env(-) VLPs modulated membrane fluidity. We have now performed a RICS analysis using Di-4-ANEPPDHQ as a lipid probe (Author response image 9). The results suggest that Env(-) VLPs do not modulate cell membrane fluidity, although we do not rule out other potential interactions with membrane proteins that might alter receptor dynamics. We appreciate the reviewer’s observation and agree that this result can be noted. However, since the main purpose of Figure 4B is to show that gp120-VLPs modulate the dynamics of CXCR4^R334X^ rather than to remark that the Env(-) VLPs also have some effects, we consider that a detailed discussion of this specific aspect would detract from the central finding and may dilute the primary narrative of the study.

Minor comments(1) It would be helpful for the reader to combine thematically or experimentally linked figures, e.g., Figures 3 and 4.(2) Figures 3 and 4 are very similar. Please unify the colours in them and the order of the panels (e.g. Figure 3 panel A shows diffusivity of CXCR4, while Figure 4 panel A shows MSI of CXCR4-R334X).

While we considered consolidating Figures 3 and 4, we believe that maintaining them as separate entities enhances conceptual clarity. Since Figure 3 establishes the baseline dynamics for wildtype CXCR4 and Figure 4 details the distinct behavior of the CXCR4^R334X^ mutant, keeping them separate allows the reader to fully appreciate the specificities of each system before making a cross-comparison.

(3) Some parts of the Discussion section could be shortened, moved to the Introduction (e.g., lines 648651), or entirely removed (e.g., lines 633-635 about GPCRs).

In accordance, the Discussion section has been reorganized and shortened to improve clarity.

(4) I suggest renaming "empty VLPs" to "Env(−) VLPs" (or similar). The name empty VLPs can mislead the reader into thinking that these are empty vesicles.

The term empty VLPs has been renamed to Env(−) VLPs throughout the manuscript to more accurately reflect their composition. Many thanks for this suggestion.

(5) Line 492 - please rephrase "...lower expression of Env..." to "...lower expression of Env or its incorporation into the VLPs...".

The sentence has been rephrased

(6) Line 527 - The data on CXCL12 modulating CXCR4-R334X dynamics and clustering are not present in Figure 4 (or any other Figure). Please add them or rephrase the sentence with an appropriate reference. Make clear which results are yours.(7) Line 532 - Do the data in the paper really support a model in which CXCL12 binds to CXCR4R334X? If not, please rephrase with an appropriate reference.

Previous studies support the association of CXCL12 with CXCR4^R334X^ (Balabanian et al. Blood, 2005; Hernandez et al. Nat Genet., 2003; Busillo & Benovic Biochim. Biophys. Acta, 2007). In fact, this receptor has been characterized as a gain-of-function variant for this ligand (McDermott et al. J. Cell. Mol. Med., 2011). The revised manuscript now includes these bibliographic references to support this commentary. In any case, our previous data indicate that CXCL12 binding does not affect CXCR4^R334X^ dynamics (García-Cuesta et al. PNAS, 2022).

(8) Line 695 - "...lipid rafts during HIV-1 (missing word?) and their ability to..." During what?

Many thanks for catching this mistake. The sentence now reads: “Although direct evidence for the internalization of CD4 and CXCR4 as complexes is lacking, their co-localization in lipid rafts during HIV-1 infection (97–99) and their ability to form heterocomplexes (22) strongly suggest they could be endocytosed together.”